# Complete Likelihood Objective for Latent Variable Models

## Abstract

In this work, we propose an alternative to the Marginal Likelihood (MaL) objective for learning representations with latent variable models, Complete Latent Likelihood (CoLLike). We analyze the objectives from the perspective of matching joint distributions. We show that MaL corresponds to a particular $KL$ divergence between some target *joint* distribution and the model joint. Furthermore, the properties of the target joint explain such major malfunctions (from the representation learning perspective) of MaL as uninformative latents (posterior collapse) and high deviation of the aggregated posterior from the prior. In the CoLLike approach, we use a sample from the prior to construct a family of target joint distributions, which properties prevent these drawbacks. We utilize the complete likelihood both to choose the target from this family and to learn the model. We confirm our analysis by experiments with low-dimensional latents, which also indicate that it is possible to achieve high-accuracy unsupervised classification using CoLLike objective.

## 1 Introduction

In the latent variable setting, the model defines a joint distribution over both observed variables $x$ and latent variables $z$, while the training data contains only observed variables. The problem can be treated as an unknown $z|x$ target conditional distribution. There are at least two possible solutions to this problem: try to come up with a meaningful target $z|x$ distribution and train the model similarly to a supervised setting, or give up and focus on matching only marginals in the $x$ domain. The latter is the choice of the MaL objective. In this work, we follow the former approach. However, instead of picking up a single target conditional we construct an entire family of possible distributions and use the model likelihood to decide which conditional to use as a target.

To construct a family of possible conditionals, we use a sample from prior of the same size as the dataset in the observed domain. All possible assignments of observed samples to latent ones span a family of empirical joint distributions. This can be represented as permutations of the latent samples. Despite the size of the permutations set being tremendous and growing as a factorial of the dataset size, the search of the permutation with the best likelihood can be done efficiently using combinatorial optimization. The resulting optimization procedure resembles the expectation maximization algorithm (Dempster et al., 1977), where expectation is replaced with the combinatorial assignment problem. Furthermore, since the proposed algorithm uses gradient-free optimization for obtaining the target distribution, the objective can be seamlessly applied to both continuous and discrete latent variables, while the discrete latents case is challenging for approaches based on the MaL(Mnih & Gregor, 2014; Mnih & Rezende, 2016; Tucker et al., 2017).

We analyze the objectives from the perspective of matching *joint* distributions. We show that MaL corresponds to a specific choice of the target $z|x$ conditional, while our approach takes into consideration family of possible conditionals. The choice of target conditional is responsible for two major failures that arise during training with the MaL objective: inability to learn informative latents, also known as "posterior collapse" (Bowman et al., 2016; Razavi et al., 2019; He et al., 2019), and divergence between the prior and the aggregated posterior (Hoffman & Johnson, 2016; Makhzani et al., 2015; Zhao et al., 2019; Kim & Mnih, 2018). These characteristics are vital for latent variable models because posterior collapse prevents learning meaningful representation and sampling from the regions of high deviation of the latent marginals are subjected to severe quality degradation

(Rosca et al., 2018). The form of the target joint also motivates the success of the complete likelihood in these challenges. Namely, the target distribution for CoLLike has high mutual information and matches prior.

We verify our analysis with experiments. In this work, we focus on low-dimensional latent variables to perform a direct comparison with the exact MaL. Models trained with CoLLike stably maintain high mutual information and low divergence from the prior. In turn, MaL inevitably leads either to posterior collapse or to a highly divergent aggregated posterior. Previously, for simple linear models, it has been shown that there is posterior collapse during the optimization of the exact likelihood Lucas et al. (2019). Our experiments demonstrate that it can as well happen with expressive models trained with exact likelihood. Along with informativeness and latent distribution matching, CoLLike indicates no degradation of likelihood compared to MaL. Furthermore, we show that CoLLike objective alone can achieve high accuracy in unsupervised classification.

We show that CoLLike unifies a range of existing approaches that lack probabilistic justification. Constrained K-means (Bennett et al., 2000), Permutation Invariant Training (Yu et al., 2017; Luo & Mesgarani, 2019), and Noise as Target Bojanowski & Joulin (2017) are among these approaches. This allows us to extend them to different factorizations of the joint and perform analysis from the probabilistic perspective. Furthermore, CoLLike bridges likelihood and optimal transport (OT) frameworks. From this perspective, the negative likelihood plays the role of both mapping from latent to visible domain and distance function.

## 2 COMPLETE LIKELIHOOD OBJECTIVE

In the regular latent variable setting, we are given a dataset $\{x_1, ..., x_N\}$ and the model $p_\theta(x, z) = p_\theta(x|z)p(z)$. The missing $z$ can be treated as the missing $p_\delta(z|x)$ part of the target joint. If we cannot come up with a reasonable $z|x$ target, we can at least match the marginals in the observed domain with $KL(p_\delta(x)||p_\theta(x))$ in hope that the model will learn an informative relation between $x$ and $z$. This is equivalent to the maximization of MaL:

$$\mathcal{L}_{MaL}(\theta) = \sum_{i=1}^{N} \log p_\theta(x_i) = \sum_{i=1}^{N} \log \int p_\theta(x_i, z)dz \tag{1}$$

Justification of the MaL comes from the equivalence of maximization of (1) and minimization of the Kullback–Leibler divergence $KL(p_\delta(x)||p_\theta(x))$ which measures the discrepancy between the target empirical data distribution $p_\delta(x)$[1] and the model distribution $p_\theta(x)$ (Murphy, 2022, 4.2.2). Note that this justifies MaL *only* for learning distributions of observed variables, not learning representations. The fact that MaL does not promote informativeness (Alemi et al., 2018) clearly shows the lack of justification of MaL for learning representation because informativeness is undoubtedly a fundamental requirement for any useful representation.

Despite the family of all possible target $p(z|x)$ distributions being tremendous, we do not need to consider it entirely. Firstly, the target distribution must be informative. Secondly, the marginal of the target joint distribution in the latent domain should match the prior $p(z)$. The fixed prior implies that the desired marginal distribution of $z$ is known. These requirements can be interpreted (Huszár, 2017) as Infomax principle (Linsker, 1988). It is not hard to get a rich family of distributions with such properties. We can obtain a collection $(z_1, ..., z_N)$ by sampling from the prior and pair this collection with the dataset $(x_1, ..., x_N)$. The pairing produce an empirical distribution. Empirical distribution attains highest possible Mutual Information (MI) under the assumption that there are no repeated values of $x$ in the dataset (see Appendix B for derivation). This ensures the first requirement. Sampling from the prior addresses the second requirement, since the collection of $z$ samples converges to $p(z)$ (Cover & Thomas, 2006, Theorem 11.2.1). However, the sampling effects can be a significant problem for high-dimensional latents. We express each pairing as some permutation $\pi$, which produces a complete collection $((x_1, z_{\pi(1)}), ..., (x_N, y_{\pi(N)}))$ and an empirical joint $p_{\delta\pi}(x, z) = p_\delta(x)p_\pi(z|x)$. Given a family of distributions, we need to decide which member of the family is our target. We propose to pick the one with the highest complete likelihood relying on the model inductive biases. For this target we then once again optimize the complete likelihood of

---

[1] We find the Greek letter $\delta$ especially suitable for data distribution because it is consonant with "data" and reflects the delta-function-like form of the empirical distribution.

the $(x_i, z_{\pi^*(i)})$ pairs with the optimal permutation $\pi^*$. These considerations lead us to the CoLLike objective:

$$\mathcal{L}_{CL}(\theta, \pi) = \sum_{i=1}^{N} \log p_\theta\left(x_i, z_{\pi(i)}\right) \tag{2}$$

which we maximize both with respect to $\theta$ and $\pi$. An alternative view on the objective can be the following: we sample $z$ values from prior and assume that they are ground truth targets for the training dataset with unknown pairing. Figure 1 depicts the main difference between the objectives: CoLLike maximizes specific points of the joint distribution, while MaL is aimed at maximization of whole lines along the joint.

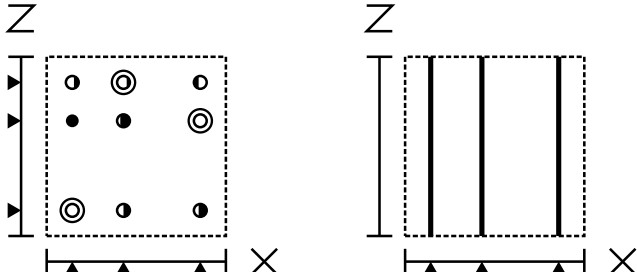

Figure 1: Illustration of the CoLLike (left) and MaL (right) objectives. Triangles depicts sample values. Filled circles represent $p_\theta(x, z)$ for all possible $(x, z)$ pairs. Double circles indicate optimal $\pi$. Bold lines and double circles are areas of the joint to be maximized.

## 3 OBJECTIVE ANALYSIS

We start our analysis by proving that MaL corresponds to the matching of a specific joint distribution and the model joint:

$$KL(p_\delta(x)p_\theta(z|x)||p_\theta(x, z)) = \mathbb{E}_{x,z \sim p_\delta(x)p_\theta(z|x)}\left[\log \frac{p_\delta(x)p_\theta(z|x)}{p_\theta(x)p_\theta(z|x)}\right] = \mathbb{E}_{x \sim p_\delta(x)}\left[\log \frac{p_\delta(x)}{p_\theta(x)}\right]$$

$$= \mathbb{E}_{x \sim p_\delta(x)}\left[\log p_\delta(x)\right] - \mathbb{E}_{x \sim p_\delta(x)}\left[\log p_\theta(x)\right] = C - \frac{1}{N}\sum_i \log p_\theta(x_i) = C - \frac{1}{N}\mathcal{L}_{MaL}(\theta)$$

where $C$ is a constant. The joint $KL$ form of the MaL brings new perspectives on the objective. It might be tempting to think about MaL as a workaround for unknown latents that allows you not to specify the target $z|x$ conditional. However, the joint form reveals that the target conditional is actually specified and equals $p_\theta(z|x)$ if we ask what distribution we want to mimic. This implies that we are aiming to keep the model posterior unchanged. In addition, the form also highlights the intimate connection between MaL and posterior.

CoLLike and a common variational (Jordan et al., 1999) approximation of MaL, Evidence Lower Bound (ELBO), can also be expressed as $KL$ divergences between joint distributions (see Table 1). We refer to Appendix A for derivation of the equivalence. Note the elegant similarity between objectives which becomes obvious in the joint $KL$ form. All divergences share the model $p_\theta(x, z)$ as the second argument, which implies that the first argument is the target *joint* distribution. For all objectives the target joint contains the data distribution $p_\delta(x)$ as a marginal in $x$ domain, thus the only difference is in the target $z|x$ conditional. Therefore, all the considered objectives belong to the family of the following form:

$$\mathcal{L}(\theta) = KL(p_\delta(x)p(z|x)||p_\theta(x, z)) = KL(p_\delta(x)p(z|x)||p_\theta(x)p_\theta(z|x))$$
$$= KL(p_\delta(x)||p_\theta(x)) + \mathbb{E}_{p_\delta(x)}\left[KL(p(z|x)||p_\theta(z|x))\right] \tag{3}$$

Since the second term in (3) is non-negative, all objectives in the family are lower bounds on the likelihood up to an additive constant. Note that the $z|x$ target conditional is used to minimize the overall divergence. This affects the second term of (3) to make the lower bound tighter.

---

[2]$q_\phi(z|x)$ is an approximate posterior distribution parametrized by $\phi$.

Table 1: Considered objectives and their joint $KL$ forms.

| | **Original Objective** | **Joint KL form** |
|---|---|---|
| **CoLLike** | $\sum_{i=1}^{N} \log p_\theta\left(x_i, z_{\pi(i)}\right)$ | $KL(p_\delta(x)p_\pi(z\|x)\|\|p_\theta(x,z))$ |
| **MaL** | $\sum_{i=1}^{N} \log p_\theta(x_i)$ | $KL(p_\delta(x)p_\theta(z\|x)\|\|p_\theta(x,z))$ |
| **ELBO**[2] | $\sum_{i=1}^{N} \mathbb{E}_{z \sim q_\phi(z\|x_i)}\left[\log \frac{p_\theta(x_i,z)}{q_\phi(z\|x_i)}\right]$ | $KL(p_\delta(x)q_\phi(z\|x)\|\|p_\theta(x,z))$ |

Despite the common traits, the objectives are different. We will highlight a few differences and go deeper in the following sections. Firstly, the target conditional for CoLLike $p_\pi(z|x)$ is empirical, while its counterparts $p_\theta(z|x)$ and $q_\phi(z|x)$ are not. Secondly, in MaL approach, we construct a particular joint distribution $p_\delta(x)p_\theta(z|x)$ and use it as a target joint, while, in CoLLike, we construct an entire family of joint distributions with desired properties. Thirdly, the target posterior is readily available in CoLLike and ELBO cases, while for MaL it could be intractable. Furthermore, CoLLike can be seamlessly applied to discrete variables, while optimization of ELBO for discrete latents is challenging (Mnih & Gregor, 2014; Mnih & Rezende, 2016; Tucker et al., 2017). Lastly, the CoLLike objective allows learning models with a reverse factorization $p_\theta(x)p_\theta(z|x)$, while MaL and ELBO do not. Reverse factorization is another inductive bias that can be useful or not. Furthermore, it can be significantly faster compared to regular factorization if $p_\theta(x)$ is assumed to be uniform and $p(z|x)$ is factorized.

## 3.1 MUTUAL INFORMATION OF THE TARGET DISTRIBUTION

Mutual Information is the key property of the joint distribution in a latent variable setting. It characterizes how dependent the observed and latent variables are. We would like to know what MI value our model is targeted at for each objective. Since our objective can be expressed as $KL$ divergence between model and target joint distributions (Table 1), we can investigate MI values for each target joint. We define MI between $x$ and $z$ under $p(x,z)$ distribution as:

$$MI(p(x,z)) = \mathbb{E}_{x,z \sim p(x,z)}\left[\log \frac{p(x,z)}{p(x)p(z)}\right] \tag{4}$$

For MaL, the MI of the target $p_\delta(x)p_\theta(z|x)$ is determined by the model's current posterior $p_\theta(z|x)$. Most models have no class preferences at initialization, which results in low MI of $p_\delta(x)p_\theta(z|x)$. Moreover, we are aimed at keeping it unchanged, since we are using the current posterior as our target posterior. So, low MI at initialization might induce learning non-meaningful factorized joint throughout the training procedure. Since for ELBO the approximate posterior aligns to the true model posterior this argument is applicable to ELBO too. Furthermore, uninformative posterior is a common problem when learning a latent variable model (Bowman et al., 2016; Alemi et al., 2018; Lucas et al., 2019; Razavi et al., 2019; He et al., 2019) known as "posterior collapse".

CoLLike target is an empirical joint distribution. It represents a deterministic mapping and has constantly high MI by construction, as shown in Appendix B. Therefore, we are aimed at mimicking a high MI distribution with our model distribution. Furthermore, CoLLike can be interpreted as some realization of InfoMax principle Huszár (2017), where prior limits the entropy and deterministic mapping maximize MI.

## 3.2 MATCHING IN THE LATENT DOMAIN

The joint form of the objectives from Table 1 is convenient for obtaining a perspective on distribution matching in the latent space. After treating $p_\delta(x)p_\theta(z|x)$ as a joint $p_{\delta\theta}(x,z)$ and rewriting the original MaL objective as:

$$KL(p_\delta(x)\|\|p_\theta(x)) = KL(p_{\delta\theta}(x,z)\|\|p_\theta(x,z)) = \mathbb{E}_{x,z \sim p_{\delta\theta}(x|z)p_{\delta\theta}(z)}\left[\log \frac{p_{\delta\theta}(x|z)p_{\delta\theta}(z)}{p_\theta(x|z)p_\theta(z)}\right]$$

$$= \mathbb{E}_{z \sim p_{\delta\theta}(z)}\left[KL\left(p_{\delta\theta}(x|z)\|\|p_\theta(x|z)\right)\right] + KL(p_{\delta\theta}(z)\|\|p_\theta(z)) \tag{5}$$

we see that matching in $x$ space requires matching in $z$ space. Namely, $KL(p_{\delta\theta}(z)||p_\theta(z)) = 0$, where $p_{\delta\theta}(z)$ is called an aggregated posterior. It signifies that even though MaL is constructed such that $z$ given $x$ conditional part of the $KL$ between joints is zero, we end up in a situation where none of the model marginals match target marginals. Moreover, the learning signal from the first term of (5) might be significantly larger compared to the second term signal if the dimensions of $x$ and $z$ differ a lot. This might lead to a sacrifice of the second divergence in favor of the first one.

Matching in a latent domain is considered as a known challenge of latent variable modelling (Hoffman & Johnson, 2016). Mismatch with prior results in unnatural samples from areas with high deviation of aggregated posterior from the prior (Rosca et al., 2018). A number of works is focused on this problem. They either utilize additional losses that penalize discrepancy between marginals (Makhzani et al., 2015; Zhao et al., 2019; Kim & Mnih, 2018) or introduce a learnable prior (Bauer & Mnih, 2019; Tomczak & Welling, 2018).

In turn, CoLLike addresses this problem by constructing a conditional, which marginal matches prior in the latent domain. Obviously, the target marginal in $x$ domain for CoLLike is always $p_\delta(x)$. In turn, the target aggregate posterior is always a sample from the prior since $p_{\delta\pi}(z) = \int_x p_\delta(x)p_\pi(z|x)dx = p_\epsilon(z)$ for all $\pi$ values, where $p_\epsilon(z)$ is the distribution of the sample produced by sampling from the prior. While it is intuitively obvious that the empirical distribution converges to the underlying distribution, one can show that $KL$ between the empirical sample and the prior converges in probability to 0 (Cover & Thomas, 2006, Theorem 11.2.1).

## 4 Algorithm

The objective (2) includes maximization with respect to two parameters: $\pi$ and $\theta$. We approach it by alternating[3] between maximization with respect to $\pi$ and $\theta$. We apply stochastic minibatch technique similar to Bojanowski & Joulin (2017), which performs maximization of both $\pi$ and $\theta$ for a minibatch instead of the entire dataset and returns the latents back to the dataset in the optimal order. Furthermore, we interpret maximization with respect to $\pi$ as a linear sum assignment problem (LAP) to utilize efficient combinatorial optimization techniques (see Appendix D for the derivation). Algorithm 1 describes the resulting stochastic optimization procedure.

---

**Algorithm 1** Stochastic optimization of CoLLike

---

**Require:** $X = (x_1, ..., x_N)$, $p_\theta(x, z) = p_\theta(x|z)p(z)$, batch size $B$, learning rate $\eta$
    Sample $Z = (z_1, ..., z_N)$ from prior ($z_i \sim p(z)$)
    **while** not converged **do**
        Sample random indices $(i_1, ..., i_B)$
        Compute matrix $\mathbf{C} \in \mathbb{R}^{B \times B}$, for which $\mathbf{C}_{q,k} = \log p_\theta(x_{i_q}, z_{i_k})$
        Compute $\pi^*$ that maximizes (2) for $(x_{i_1}, ..., x_{i_B}), (z_{i_1}, ..., z_{i_B})$ by applying LAP solver to $\mathbf{C}$
        $\theta \leftarrow \theta - \eta \nabla_\theta \mathcal{L}_{CL}(\theta, \pi^*)$
        Put $(z_{i_1}, ..., z_{i_B})$ back into $Z$ in the optimal order $(z_{\pi^*(i_1)}, ..., z_{\pi^*(i_B)})$
    **end while**

---

The core of the algorithm is in computation of the matrix $\mathbf{C}$ and optimal permutation $\pi^*$. Both parts are potentially computationally intense and challenging.

Computation of the matrix $\mathbf{C}$ requires $B^2$ forward passes. Note that backward passes are not required for this step, hence, memory requirements are mild. Furthermore, in this work we focus on low-dimensional discrete latents. Assuming the number of categories $K$ of the latent variable $z$ is lower than $B$, the number of *all* possible values of $z$ in $\log p_\theta(x_{i_q}, z)$ equals $K$ instead of $B$. Thus, we can calculate all needed values of $\mathbf{C}$ using only $K \cdot B$ forward passes instead of $B^2$. In the supervised case, we only need the $\theta$ update part of the entire algorithm, which requires $B$ forward passes and $B$ backward passes. As a rough estimate, we can assume that forward and backward passes take the same time, CoLLike will then require $K/2$ more compute time compared to the supervised setting.

---

[3]This is similar to EM algorithm Dempster et al. (1977) in the sense that EM alternates between maximization of the lower bound tightness (expectation step) and maximization of the the resulting tight bound (maximization step).

To find $\pi^*$ we use Hungarian algorithm Kuhn (1955) as a LAP solver. The algorithm requires the cost matrix $\mathbf{C} \in \mathbb{R}^{B \times B}$ as input to produce the optimal permutation $\pi^*$ in the form of optimal ordering of $(z_{i_1}, ..., z_{i_B})$ indices. The complexity of the algorithm is $\mathcal{O}(B^3)$. The complexity of the LAP solver potentially limits the applicability of CoLLike to large batch sizes. However, for batch sizes regularly used in practice, solving LAP results in only a minor increase in the overall computation time. For instance, in our experiments, we used batch size 64. Solving the LAP took orders of magnitude less time compared even to the supervised setting. See Appendix E for detailed timings. We also highlight that optimization with large batches is not only challenging but also could significantly reduce generalization (Xing et al., 2018; You et al., 2019; 2020). However, as gracefully shown by Huszár (2017), this kind of minibatch combinatorial optimization provides only locally optimal solutions. Nevertheless, the size of the gap between local global optimum is still to be determined.

The result of Algorithm 1 is a trained model. However, we are interested in the posterior $p_\theta(z|x)$. For low-dimensional categorical $z$, we can exactly compute the posterior using Bayes rule $p_\theta(z|x) = p_\theta(x, z)/p_\theta(x)$ since $p_\theta(x) = \sum_i p_\theta(x, z = i)$ is tractable. For other cases, we can fit an approximate posterior using regular variational techniques. We can also use CoLLike objective to obtain estimates of $z$ values. After training model with CoLLike objective we have $X$ and $Z$ arrays that are matched. We add new $x$ samples to $X$ and sample extra $z$ values from the prior to extend $Z$. The inference can then be performed by optimizing 2 with respect to $\pi$.

## 5 CONNECTIONS

Connections with existing techniques not only give alternative perspectives on CoLLike objective, but also provide probabilistic grounding to some existing algorithms. Many well-known objectives actually use CoLLike while being motivated as an ad-hoc empirical risk minimization. We show that these objectives not only seem reasonable but are also probabilistically motivated.

While traditional K-means algorithm (MacQueen, 1967; Lloyd, 1982) has a probabilistic grounds (Murphy, 2022, 21.4.1.1), its constrained counterpart (Bennett et al., 2000) lacks probabilistic justification. Constrained K-means is equivalent to CoLLike under factorized Gaussian $p_\theta(x|z)$ and uniform categorical $p(z)$, which has a number of states equal to the number of clusters. This connection allows extending the constrained K-means approach to different generative distributions and priors. Nevertheless, a probabilistic interpretation is present in Jitta & Klami (2018), however, the choice of the complete likelihood as an objective is not explained.

Permutation Invariant Training (PIT) (Yu et al., 2017; Luo & Mesgarani, 2019) used in source separation solutions can also be expressed as CoLLike objective. For instance, in cocktail part problem, we want to separate a mixture of $K$ sources. During training, we have $K$ isolated mixture components and a network that produces $K$ estimates of the components based on a single mixture. We don't know which network output corresponds to which source and we pick a permutation that produces minimal total mismatch between outputs and sources. This procedure corresponds to training a latent variable model with CoLLike objective, where a categorical latent variable of dimension $K$ determines the source identity. In this setting, we treat mixture components as samples in the dataset.

The closest predecessor of the CoLLike is Noise As Target (NAT) (Bojanowski & Joulin, 2017). This is an unsupervised approach to learn an image encoder. In this approach, the representations produced by a network are assigned to a fixed collection of vectors sampled from the uniform distribution on a sphere. After this, the network parameters are adjusted to make encodings closer to the assigned vectors. This approach is equivalent to CoLLike with reverse model factorization $p_\theta(x, z) = p_\theta(z|x)p(x)$ and factorized Gaussian $p_\theta(z|x)$. Another approaches that obtain clear probabilistic interpretation using CoLLike include: Sinkhorn Autoencoders (Patrini et al., 2019), simultaneous clustering and representation learning (Asano et al., 2020), and (Jeong & Song, 2019).

Bojanowski & Joulin (2017) noticed that NAT objective has Optimal Transport (OT) roots. OT framework can be used to measure discrepancy between distributions. Particularly, for a given non-negative cost function $c$ the optimal transport distance between distributions $p_\delta$ and $p_\epsilon$ is defined as

$$OT(p_\delta, p_\epsilon) = \min_{\gamma \in \Gamma(p_\delta, p_\epsilon)} \mathbb{E}_{x, z \sim \gamma(x, z)} \left[ c(x, z) \right]$$

Table 2: Results for tractable categorical latents. MNIST, CIFAR – BPD; AG News – NLL.

| Dataset | Objective | Accuracy ↑ | NLL/BPD ↓ | Agg. KL ↓ | MI ↑ |
|---------|-----------|-----------|-----------|-----------|------|
| CIFAR | CoLLike | 14.5 | 3.45 | 0.01 | 2.20 |
| CIFAR | MaL | 14.0 | 3.46 | 0.74 | 1.50 |
| MNIST | CoLLike | 14.1 | 1.27 | 0.01 | 1.95 |
| MNIST | MaL | 12.5 | 1.29 | 1.61 | 0.58 |
| AG News | CoLLike | 82.1 | 250.79 | 0.00 | 1.32 |
| AG News | MaL | 31.6 | 249.73 | 0.00 | 0.00 |

where $\Gamma(p_\delta, p_\epsilon)$ is the set of all joint distributions on $x$ and $z$ with marginals $p_\delta(x)$ and $p_\epsilon(z)$ respectively. Furthermore, if we use a parametric model $p_\theta$ in place of $p_\epsilon$ we can fit it by minimizing the distance. Note that in this case we minimize the function that already has a $\min$ function inside.

When both $p_\delta$ and $p_\epsilon$ are empirical, the search space $\Gamma$ becomes countable and finite. Now it contains only pairings between points in $p_\delta$ and points in $p_\epsilon$. Given an arbitrary initial pairing, we can express all other pairings through permutation applied to either $x$ or $z$. In this case, the cost becomes

$$OT(p_\delta, p_\epsilon) = \min_{\pi \in \Pi} \sum_i c\left(x_i, z_{\pi(i)}\right)$$

where $\Pi$ is the set of all permutation functions. This expression is almost the CoLLike objective (2). Choosing the cost function $c$ to be $-\log p_\theta(x, z)$ and switching to maximization make them equivalent[4]. Thus, CoLLike bridges maximum likelihood methods with OT. This connection allows bringing latest developments in OT to improve likelihood-based methods. Furthermore, in Appendix C, we provide an example of the equivalence between CoLLike and Wasserstein distance. In the case, the model's complete likelihood plays the roles of both a mapping from $z$ to $x$ domain and a distance metric.

## 6 EXPERIMENTS

In this work, we focus on low-dimensional discrete latents. This type of latent variables allows to perform direct comparison with the exact likelihood. Furthermore, we emphasize our focus on learning useful $z|x$ instead of simplifying the model with factorized $x|z$ conditional.

Models with tractable likelihood are perfect for comparing likelihood-based algorithms because they remove the problem of the likelihood estimation precision. For this type of models, all quantities of interest can be computed exactly. Moreover, tractable likelihood allows comparing CoLLike directly with MaL instead of its approximations like ELBO.

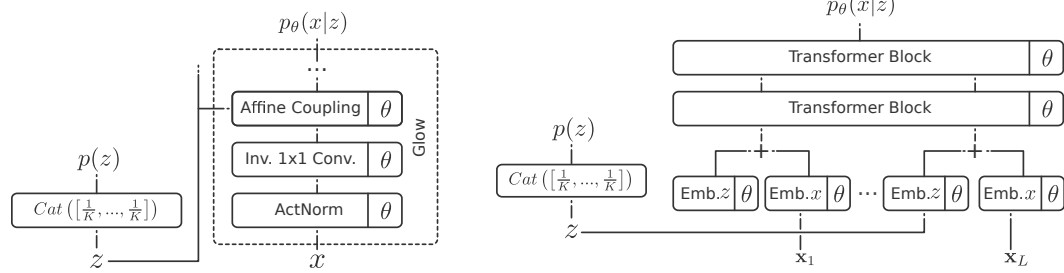

Figure 2: Architectures used for image (left) and text (right) domains. For image domain, the three flow blocks are repeated 21 (CIFAR) and 14 (MNIST) times. Every coupling block is conditioned on $z$.

We use MNIST (LeCun et al., 1998) and CIFAR (Krizhevsky, 2009) datasets for image modality and AG News (Zhang et al., 2015) for text domain. All these datasets are equipped with class labels.

---

[4]A cautious reader might note that for continuous variables non-negativity of $\log p_\theta$ can be violated, however, all model densities used in practice are finite and the corresponding cost can be made positive just by an additive constant which does not change the optimization problem.

For images, we train a Glow-like normalizing flow conditioned on a discrete latent variable with 10 categories through all coupling layers. For text we use a Transformer Language Model conditioned on a discrete latent variable with 4 categories using additive embedding for all tokens. The size of the discrete variables is equal to the number of classes in the underlying dataset. Small number of categories allows to compute exact marginal likelihood value and speed up computation of the cost matrix $\mathbf{C}$ in Algorithm 1. The schematic representations of the architectures are provided in Figure 2.

Table 2 presents the results of training the latent variable models for CoLLike and MaL objectives averaged across 4 runs. Both objectives exhibit similar performance in terms of likelihood across datasets. However, other characteristics vary.

MI is high for CoLLike objective on every dataset. Furthermore, it attains approximately maximal value for AG News and CIFAR. MI for MaL objective ranges from zero to values significantly lower than those of CoLLike. Zero MI indicates posterior collapse cases, which are mainly observed in ELBO optimization and recently discovered by Lucas et al. (2019) for MaL applied to simple linear models. This experiment indicates important observation: posterior collapse can as well happen in deep latent variable models during optimization of exact MaL despite usually being corresponded to the structure of ELBO. Importantly, for the MNIST dataset, half of the experiments exhibits posterior collapse.

CoLLike exhibits near-zero aggregated KL for all experiments. It implies that the model joint marginal in the latent domain perfectly matches the prior. For MaL, aggregated KL is zero only for AG News dataset which also has uninformative factorized joint. For other datasets, aggregated posterior significantly deviates from the prior. We also note that for MNIST dataset, MaL puts all probability mass to a single category in half of the runs.

To estimate the quality of unsupervised classification, we perform the optimal assignment of latent categories to classes. For all cases except CoLLike objective on AG News dataset, the quality of the unsupervised classification is similar and is low. On AG News the unsupervised accuracy is exceptionally good. However, the variance of the proposed solution is relatively high. The standard deviation of the accuracy across 4 runs is 5.4 with the highest value of 87.1 and the lowest of 73.3. In the following section, we show that it is possible to achieve significantly higher unsupervised accuracy and lower variance by latent variable ensembling.

Overall, CoLLike clearly outperforms MaL in the tractable likelihood setting. Moreover, it shows high unsupervised classification accuracy for text modality. For MaL, experiments depict a variety of possible failures from posterior collapse to degenerate aggregated posterior, which extends findings of (Lucas et al., 2019) to expressive models and exact likelihood. However, despite CoLLike producing informative latents in terms of MI, unsupervised classification might be challenging even in these cases. We believe that the key to high-performance unsupervised classification should be in the right inductive biases in conditioning and probabilistic model type.

## 6.1 LATENT ENSEMBLING

To reduce the high variance of CoLLike unsupervised classification accuracy and increase its accuracy we propose to perform ensembling of multiple models trained on the same data but using different seeds at initialization. Although there is no correspondence between labels for latent variable models, we can try to find the labels assignment based on the agreement between them. This approach is motivated by direct cluster ensembling (Boongoen & Iam-on, 2018). The agreement between two labels of different ensemble members is the number of intersecting samples with those labels. To align the latents we iteratively find the assignment with the highest intersection between labels. Finally, we find the assignment between aligned latents and ground truth labels.

In our experiments, we use 8 models per ensemble and train 4 independent ensembles. The simplest ensembling method is averaging of the predictions. It increases the mean unsupervised accuracy from 82.1 to 84.5 and reduces the standard deviation from 5.4 to 1.7. We further significantly improve these results by utilizing the agreement score, which is also used for alignment of the labels. We pick top-k models with highest maximum coherence across other models in the ensemble. Averaging predictions of those top-k models further increases accuracy to **86.6** and lowers the standard deviation to **0.2**.

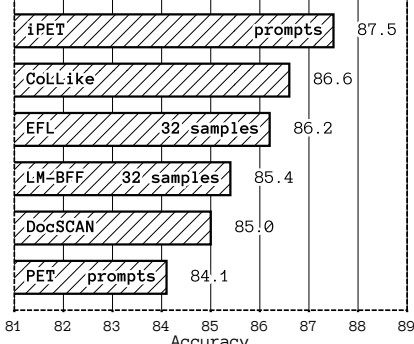 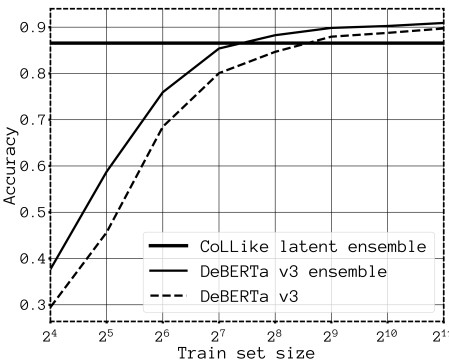

(a) CoLLike ensemble vs. unsupervised and semi-supervised approaches.

(b) Supervised DeBERTa v3 base vs. CoLLike ensemble.

Figure 3: Comparison of ensembled CoLLike with supervised (a) and unsupervised/few-shot methods(b).

We compare CoLLike results with the following unsupervised and supervised approaches: PET and iPET (Schick & Schütze, 2021), EFL (Wang et al., 2021), LM-BFF (Gao et al., 2021), DocSCAN (Stammbach & Ash, 2021). DocSCAN is purely unsupervised, while other approaches rely on engineering multiple textual descriptions of classes (prompts) or labeled data. All methods use heavy pre-trained Transformers (Vaswani et al., 2017) as an initialization, while in CoLLike we use small 2-layer Transformer with random initialization. Figure 3a presents the comparison of the methods. CoLLike clearly outperforms both unsupervised and most of the supervised methods. To determine how much training data we need without, possibly laborious, prompt engineering we use DeBERTa v3 (He et al., 2021). We vary training set sizes from 32 to 2048 and apply additional ensembling of 8 models with different initializations and train-validation splits. Figure 3b reveals that CoLLike can be a better alternative to labeling more than a hundred samples, which, in turn, requires an extensive data analysis. Besides, note the high difference between the ensemble and the single model for small dataset sizes in a supervised setting, which is an interesting result by itself.

# 7 DISCUSSION AND FUTURE WORK

In this work, we propose to switch from the MaL paradigm of matching only marginals in the observed domain to CoLLike paradigm of finding an exact target joint by selection from a family of joints with desirable properties. Furthermore, we show that matching of marginals utilized by MaL corresponds to a specific choice of target joint, which motivates such failures as posterior collapse and divergence between target and model marginals in the latent domain. We experimentally show the ability of CoLLike to learn useful representations. Connection of CoLLike with OT allows to borrow techniques from the latter. For instance, Sinkhorn Relaxation (Cuturi, 2013) can be used to speed up the assignment problem. Investigation of alternatives to complete likelihood for target selection is of special interest. The right inductive biases for inducing useful properties using CoLLike are still to be discovered, at least until we want to get the desired without specifying what we want. We believe that the further extension of CoLLike to high-dimensional latents would be exciting and challenging. Other lines of research can be devoted to the application of other divergences to the constructed family of joint target candidates and extension of CoLLike to learnable priors.

# 8 REPRODUCIBILITY

To promote reproducibility we open-source our code *the link is hidden for double-blind review, check the supplementary materials*. Furthermore, we describe details of flow architectures in Appendix F.1 and Transformers in Appendix F.2. Along with models, we describe details of the training

procedures and data pre-processing. We also devote special attention to setting all necessary seeds, including CUDA, and to removing stochasticity from the BPE tokenizer.

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

## A    DERIVATION OF KL FORMS OF THE CONSIDERED OBJECTIVES

Equivalence between CoLLike objective (2) and its $KL$ divergence form from Table 1 can be derived as follows:

$$
\begin{aligned}
KL(p_\delta(x)p_\pi(z|x)||p_\theta(x,z)) &= \mathbb{E}_{x,z\sim p_\delta(x)p_\pi(z|x)}\left[\log\frac{p_\delta(x)p_\pi(z|x)}{p_\theta(x,z)}\right] \\
&= \mathbb{E}_{x,z\sim p_\delta(x)p_\pi(z|x)}\left[\log p_\delta(x)p_\pi(z|x) - \log p_\theta(x,z)\right] \quad (6)\\
&= C - \mathbb{E}_{x,z\sim p_\delta(x)p_\pi(z|x)}\left[\log p_\theta(x,z)\right] \\
&= C - \frac{1}{N}\sum_{i=1}^{N}\log p_\theta(x_i, z_{\pi(i)}) \\
&= C - \frac{1}{N}\mathcal{L}_{CLL}(\theta,\pi) \quad (7)
\end{aligned}
$$

where the first term in (6) is treated as constant with the assumption that all samples from $p_\delta(x)$ take distinct values, which is reasonable for such high-dimensional objects as images, texts, and sounds. Thus, the $KL$ form of the objective is equivalent to the CoLLike objective up to a multiplicative factor and an additive term. For proof of the constancy see Appendix B.

The derivation of equivalence between (1) and its $KL$ from Table 1 is as follows

$$
\begin{aligned}
KL(p_\delta(x)p_\theta(z|x)||p_\theta(x,z)) &= \mathbb{E}_{x,z\sim p_\delta(x)p_\theta(z|x)}\left[\log\frac{p_\delta(x)p_\theta(z|x)}{p_\theta(x)p_\theta(z|x)}\right] \\
&= \mathbb{E}_{x\sim p_\delta(x)}\left[\log\frac{p_\delta(x)}{p_\theta(x)}\right] \\
&= \mathbb{E}_{x\sim p_\delta(x)}\left[\log p_\delta(x)\right] - \mathbb{E}_{x\sim p_\delta(x)}\left[\log p_\theta(x)\right] \\
&= C - \frac{1}{N}\sum_i \log p_\theta(x_i) \\
&= C - \frac{1}{N}\mathcal{L}_{MaL}(\theta)
\end{aligned}
$$

The $KL$ form of ELBO objective from Table 1 can be found in many works Zhao et al. (2019); Kingma & Welling (2019), however, we provide a derivation here to make the paper self-contained.

$$
\begin{aligned}
KL(p_\delta(x)q_\phi(z|x)||p_\theta(x,z)) &= \mathbb{E}_{x,z\sim p_\delta(x)q_\phi(z|x)}\left[\log\frac{p_\delta(x)q_\phi(z|x)}{p_\theta(x,z)}\right] \\
&= \mathbb{E}_{x\sim p_\delta(x)}\left[\log p_\delta(x)\right] + \mathbb{E}_{x,z\sim p_\delta(x)q_\phi(z|x)}\left[\log\frac{q_\phi(z|x)}{p_\theta(x,z)}\right] \\
&= C - \mathbb{E}_{x\sim p_\delta(x)}\mathbb{E}_{z\sim q_\phi(z|x)}\left[\log\frac{p_\theta(x,z)}{q_\phi(z|x)}\right] \\
&= C - \frac{1}{N}\sum_i\mathcal{L}_{ELBO}(x_i,\phi,\theta)
\end{aligned}
$$

## B    ENTROPY AND MUTUAL INFORMATION OF EMPIRICAL JOINT

In this appendix we derive some useful properties of the empirical joint distributions produced by sampling from the prior. The joint distribution $p_\delta(x)p_\pi(z|x)$ depends on $\pi$. We focus on how $\pi$ influences such distribution characteristics as entropy and mutual information.

Consider a joint distribution over discrete $x$ and $z$. This kind of distribution can be visualized as a table, such as depicted in Figure 4. If there are multiple samples taking the same value both in $x$

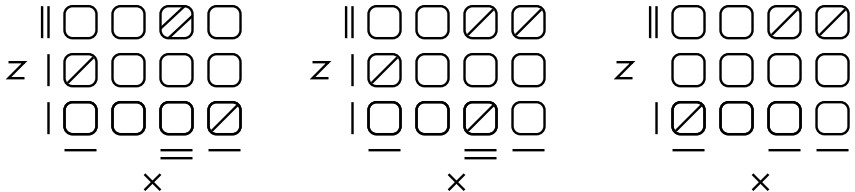

Figure 4: Example of three joint distributions with discrete $x$ and $z$. Lines on the left and in the bottom depict the number of samples from empirical marginals with the corresponding value of the random variable. Squares reflect the joint probability value. Each line crossing the square corresponds to $1/N$ probability added to the corresponding $(x, z)$ random variable pair.

and $z$ domain, the permutation can change the entropy of the joint. For instance, for the distribution on the left, the entropy $H(p_\delta(x)p_\pi(z|x)) = -(\frac{1}{4}\log\frac{1}{4} + \frac{1}{4}\log\frac{1}{4} + \frac{1}{2}\log\frac{1}{2}) \approx 1.04$ nats. For the distribution in the center, the entropy $H(p_\delta(x)p_\pi(z|x)) = -(4 \cdot \frac{1}{4}\log\frac{1}{4}) \approx 1.39$ nats. So, depending on $\pi$ we might end up with more and less entropic distributions.

However, when we restrict any empirical marginal to take only distinct values, like in the right part of Figure 4, the situation changes. Namely, each distinct pair $(x, z)$ can be chosen at most once, because to choose it twice we need a duplicate sample in both domains. This can be verified using the right part of Figure 4. Just try to construct a joint with some square having grater than one line assuming that each value $x$ marginal has only one line. Moreover, for $x$, this is a reasonable assumption since usually the domain of $x$ is high-dimensional. For the case, the joint will contain $N$ non-zero points each with probability $1/N$. Thus, the entropy of the empirical distribution is equal $-\sum_{n=1}^{N}\frac{1}{N}\log\frac{1}{N} = \log N$.

The observation above allows to easily derive mutual information of the empirical joint. Mutual information is defined as

$$MI(p(x, z)) = \mathbb{E}_{x,z \sim p(x,z)}\log\frac{p(x, z)}{p(x)p(z)} = \mathbb{E}_{x,z \sim p(x,z)}\log\frac{p(z|x)}{p(z)}$$

Under the assumption that $p_\delta(x)$ contains only distinct elements the conditional $p_{\delta\pi}(z|x) \equiv 1$ for all values $x$ from the $p_\delta(x)$. So, choosing $x$ uniquely determines the value of $z$, as can be seen from the right part of Figure 4. Then the mutual information is given by

$$MI(p_\delta(x)p_\pi(z|x)) = \mathbb{E}_{x,z \sim p_\delta(x)p_\pi(z|x)}\log\frac{p_{\delta\pi}(z|x)}{p_{\delta\pi}(z)} = \mathbb{E}_{z \sim p_{\delta\pi}(z)}\log\frac{1}{p_{\delta\pi}(z)} = H(p_{\delta\pi}(z)) \quad (8)$$

So, the mutual information is always equal to the entropy of the empirical prior. It is possible to show that the value is the maximum possible one. This becomes obvious from the entropic factorization of the mutual information

$$MI(p(x, z)) = H(p(z)) - \mathbb{E}_{x \sim p(x)}\left[H(p(z|x))\right] \quad (9)$$

Since the entropy is non-negative, the mutual information can be decreased only through the second term of (9), which equals 0 because $z$ value is completely determined by $x$.

When we try to extend the observations above to continuous cases we face the following challenge: empirical distribution has infinite values at the sample points. This drives the differential entropy as well as mutual information to infinity. However, adding noise to the empirical distribution solves this problem. Adding uniform noise with the interval smaller than the precision of the floating point makes the entropy finite and constant with respect to $\pi$. One can show that the resulting mutual information of the empirical joint also equals $\log N$.

## C  WASSERSTEIN DISTANCE AND CoLLIKE

Optimal Transport cost becomes Wasserstein distance when $c$ is a metric. A very illustrative example from this family is equality of Wasserstein-2 ($c$ is the Euclidean distance) and CoLLike for some setups. Specifically, the following objective can be produced both by Wasserstein distance and CoLLike

$$\mathcal{L}_W(\theta) = \min_{\pi \in \Pi} \sum_i \left( x_i - f_\theta(z_{\pi(i)}) \right)^2$$

To get this objective from OT perspective we define the model distribution to be produced by passing a fixed sample from prior through a deterministic decoder $f_\theta(z)$. The result is an empiric distribution in $x$ domain. Wasserstein distance between two empiric distributions is determined by optimal pairing between points from data distribution $p_\delta(x)$ and model distribution spanned by empiric latents. The same objective is produced by factorized Gaussian $p_\theta(x|z)$ and uniform prior $p(z)$.

This connection demonstrates that the model $p_\theta(x|z)$ defines both mapping from $z$ to $x$ domain and "topology" of the $x$ space (how we measure distance between objects). However, approaches based on the Wasserstein distance are limited to continuous variables, while CoLLike is applicable both for discrete and continuous domains. Moreover, CoLLike provides a probabilistic basis for the choice of the cost function.

## D  OPTIMAL PAIRING BY COMBINATORIAL OPTIMIZATION

Having at hand log-likelihood values for all possible $x_i$ $z_j$ pairs, we are ready to find the optimal permutation. A naive way to do so is to evaluate the sum 2 for every possible permutation $\pi$. Despite we need only to sum different pre-computed values, the search space for $\pi$ is tremendous $N!$. However, we can cast this problem to a combinatorial optimization one. Following Papadimitriou & Steiglitz (1982), the assignment problem is stated as follows

$$\begin{aligned} \text{minimize} \quad & \sum_{i,j} c_{i,j} a_{i,j} \\ \text{subjected to} \quad & \sum_i a_{i,j} = 1 \quad j = 0, ..., N \\ & \sum_j a_{i,j} = 1 \quad i = 0, ..., N \\ & a_{i,j} \in \{0, 1\} \end{aligned}$$

where $c_{i,j}$ is the cost of picking the element $i, j$ and $a_{i,j}$ is the indicator variable. The constrains of this problem define a set of permutation matrices. By choosing the cost to be negative log-likelihood and replacing indicator variable with permutation we end up with CoLLike objective. This combinatorial optimization problem can be solved efficiently with Hungarian algorithm Kuhn (1955) with complexity of $\mathcal{O}(N^3)$.

## E  THE COMPLEXITY OF THE LAP SOLVER

Figure 5 depicts dependency between LAP problem size and time consumed by Hungarian algorithm to solve the problem. The input to the algorithm is a matrix $\mathbb{C} \in \mathbb{R}^{B \times B}$, where $B$ is the size of the problem.

## F  MODELS DESCRIPTION

### F.1  NORMALIZING FLOWS

We use Glow-like normalizing flow for all image experiments. We choose the learning rate by starting from $1e^{-2}$ and gradually decrease it until there is no instabilities during training. No extensive learning rate search was done. Below we provide details on the model parameters.

CIFAR model:

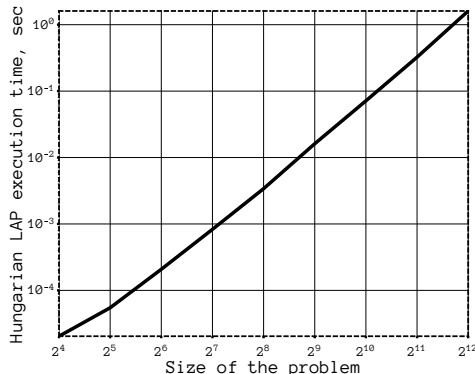

Figure 5: Time to solve LAP with Hungarian algorithm for different sizes of the problem.

- architecture: Glow
- number of flows per scale: 7
- flow coupling: affine
- coupling net: ResNet, 3 blocks, hidden size 96
- permutation flow: invertible $1 \times 1$ convolution with lower-upper factorization
- normalization flow: ActNorm
- number of scales: 3
- scale factor: 2 height, 2 width

MNIST model (same as CIFAR except the number of scales):

- architecture: Glow
- number of flows per scale: 7
- flow coupling: affine
- coupling net: ResNet, 3 blocks, hidden size 96
- permutation flow: invertible $1 \times 1$ convolution with lower-upper factorization
- normalization flow: ActNorm
- number of scales: 2
- scale factor: 2 height, 2 width

During training we use marginal likelihood to validate both MaL and CoLLike. We did not search over possible optimizers, and use Adam (Kingma & Ba, 2015) with default parameters. The validation split is chosen to be 0.05 because no significant variations of the likelihood were observed during training. To summarize, we use the following parameters both for CoLLike and MaL:

- epochs: 256
- learning rate: $5e^{-5}$ - MNIST; $2e^{-4}$ - CIFAR
- batch size: 64
- validation part of the training set: 0.05
- validation criterion: marginal likelihood
- optimizer: Adam, $\beta = (0.9, 0.999); \epsilon = 1e^{-8}$

As data pre-processing step, we used only dequantization with uniform noise, with the range equal to the quantization step.

## F.2 TRANSFORMER

We used simple two-layer transformer across our experiments. The model description:

- number of layers: 2
- hidden size: 128
- feedforward dimension: 128
- embedding dimension: 128
- number of attention heads: 4
- number of embeddings: 4000

The training details are similar to CIFAR configuration:

- epochs: 256
- learning rate: $2e^{-4}$ - CIFAR
- batch size: 64
- validation part of the training set: 0.05
- validation criterion: marginal likelihood
- optimizer: Adam, $\beta = (0.9, 0.999); \epsilon = 1e^{-8}$

In pre-processing step, we truncate the sequences longer than 192 tokens. Truncation affects less than $0.3\%$ of the samples. Nevertheless, the tokenizer is trained on the full-length sequences. Data pre-processing can be summarized as follows:

- maximum length truncation: 192
- BPE tokenization
- vocabulary size: 4000

