# OpenReview forum: "Complete Likelihood Objective for Latent Variable Models"
_ICLR.cc/2023/Conference — Submitted to ICLR 2023_

### Official Review · Reviewer_vRN3 · 2022-10-21

**Confidence:** 4
**Correctness:** 2
**Technical Novelty And Significance:** 1
**Empirical Novelty And Significance:** 1
**Recommendation:** 1

**Clarity, Quality, Novelty And Reproducibility:**

While the claim of the paper is clear, the proposed algorithm is a naive variant of MLE, and is not supported by convincing reasons.

**Strength And Weaknesses:**


There are a number of controversial heuristics unjustified in my view.

1. One basic principle of the MLE is its (asymptotic) consistency (which is why in the first place people want to do ML estimation). What can be said about the proposed algorithm? Is it consistent at least?

2. The proposed algorithm is not a strict generalization of K-means, since empirical distribution of true labels does not necessarily match the true prior. Impact of such sampling errors should be considered (e.g., when clusters are very well-separated).

3. For AG News data, the huge gap in Accuracy is suspicious. More thorough and detailed discussion on this phenomenon (on why it happens) should follow.



Other weaknesses

- Discussions in Section 5 are reasonable, but they are sort of trivial and do not add much value to the paper in my opinion.

- What can be done if the prior of latent variables are not known in advance?

- In the experiments, it is not clear what NLL / Agg. KL stand for (and also Accuracy is not precisely defined).

- It is unusual to use $-\log$ as a distance measure for OT. So why this should be better than using a typical Euclidean distance measure?


**Summary Of The Paper:**

The submission proposes a Complete Latent Likelihood (CoLLike) objective, which is given as (3), as an alternative to commonly used log-likelihood objectives (which the authors call Marginalized Likelihood, MaL) for learning latent variable models. In essence, when the prior over latent variables are known or given, the proposed method generates i.i.d. latent variables from the prior, and find a permutation between the generated latent variables and observed samples.

The motivation comes from scenarios where computing a marginalized probability by integrating over latent variables can be intractable, typically required in EM style algorithms. Instead, by finding the optimal pairing of observed samples and (generated and fixed) latent variables, which can basically be reduced to a bipartite matching (or assignment) problem, authors could avoid the intractable integration step.

Experiments are performed on datasets with relatively small number of latent classes, and compare the performance of CoLLike and MaL.

**Summary Of The Review:**

I do not think the paper contains enough contribution or originality to be published.

---

> ### Author Response · Authors · 2022-11-19
> **Response**
>
> Thank you for the review.
>
>
> > One basic principle of the MLE is its (asymptotic) consistency (which is why in the first place people want to do ML estimation). What can be said about the proposed algorithm? Is it consistent at least?
>
> Following the introductory book on machine learning [1, sec. 5.3.2]:
>
> ```Although consistency is a desirable property, it is of somewhat limited usefulness in practice since most real datasets do not come from our chosen model family (i.e., there is no θ ∗ such that p(·|θ ∗ ) generates the observed data D). In practice, it is more useful to find estimators that minimize some discrepancy measure between the empirical distribution p D (x|D) and the estimated distribution p(x| θ̂). If we use KL divergence as our discrepancy measure, our estimate becomes the MLE.```
>
> It would be beautiful to prove consistency or the reverse, but we do not consider addressing this question at this time.
>
>
>
> > The proposed algorithm is not a strict generalization of K-means, since empirical distribution of true labels does not necessarily match the true prior.
>
> Yes, but this observation comes from the reviewer but not from the paper, we clearly state that our algorithm is a generalization of **Constraint K-Means**.
>
> > Impact of such sampling errors should be considered (e.g., when clusters are very well-separated).
>
> This impact is considered by the aggregated $KL$ divergence. For low-dimensional models it is negligible. However, this can be a significant problem for high-dimensional $z$. We added this consideration to the objective section.
>
> > For AG News data, the huge gap in Accuracy is suspicious. More thorough and detailed discussion on this phenomenon (on why it happens) should follow.
>
> It is not suspicious, this follows from the MI.
>
> > What can be done if the prior of latent variables are not known in advance?
>
> We do not address this case in the paper. A learnable prior should be introduced, this is a possible direction for future work.
>
> > In the experiments, it is not clear what NLL / Agg. KL stand for (and also Accuracy is not precisely defined).
>
> NLL is a standard notation used in all papers devoted to probabilistic generative models. Agg. KL is just an abbreviation for Aggregated KL, which is described in Section `3.2`. Accuracy is defined by the following statement: `To estimate unsupervised classification quality we perform the optimal assignment of latent categories to classes`. We do not see any ambiguity here.
>
> > It is unusual to use $\log$ as a distance measure for OT. So why this should be better than using a typical Euclidean distance measure?
>
> We stated in the body that it might result in exactly Euclidean distance: `we provide an example of the equivalence between CoLLike and Wasserstein distance`.
>
>
> > Discussions in Section 5 are reasonable, but they are sort of trivial and do not add much value to the paper in my opinion.
> > While the claim of the paper is clear, the proposed algorithm is a naive variant of MLE, and is not supported by convincing reasons.
>
> It follows from these two statements that all objectives from Section 5 can be summarized as "naive variants of MLE" which are "not supported by convincing reasons".
>
> We believe that the summary of the review is mainly motivated by the wrong assumption that MaL is somehow justified for representation learning.
>
> > the proposed algorithm is a naive variant of MLE
>
> MLE is an algorithm for matching distributions of the observed variables. Therefore, we assume that you are referring to Marginal Likelihood by using MLE term. Naïve solution implies that some complicated parts of the problem are discarded. From this perspective, Marginal Likelihood discards the fundamental part of any representation learning objective - informativeness. What kind of "naiveness" are you referring to?
>
> > and is not supported by convincing reasons
>
> We special efforts to make the motivation of the objective clear. We highlight that Marginal Likelihood is not supported for representation learning. We also reflect it in Section 2.
>
>
> [1] Kevin P. Murphy.Probabilistic Machine Learning: An introduction. MIT  Press, 2022. probml.ai

---

### Official Review · Reviewer_wzh3 · 2022-10-24

**Confidence:** 3
**Correctness:** 2
**Technical Novelty And Significance:** 2
**Empirical Novelty And Significance:** 2
**Recommendation:** 3

**Clarity, Quality, Novelty And Reproducibility:**

The authors provided their code as supplementary material, and the quality and the novelty are fair.

For the clarity of the paper, here are some questions to the authors.

- In the beginning of Section 2, the authors should mention that x is an input and y is a label to be predicted, even though it is convention. Or, am I wrong?

- I wonder why the authors represent the latent variable z as “label”. Does the latent z is limited to some attributes, for example? Can’t it be in any shape? It would be better to provide a graphical notation of x, y, and z, together.

- Sampling from the prior ensures the first requirement, … -> probably (x_N, y_{pi(N)}) ?

- Figure 1 is very hard to follow. What are the circles indicates? Where is permutation pi in the figure?

- The python-like pseudo-code should be written with more consistency.

- The complexity of CoLLike seems to be problem. Even though the authors utilized the combinatorial linear sum assignment problem with complexity of O(N^3), the problem is that N is the number of data instances which can be a huge number. Could you compare the wall-clock runtime of CoLLike against the baselines? Also, I’m curious about the effect of additional techniques: minibatch assignment, low-dimensional discrete latents, and factorized conditional. How much do they reduce the wall clock runtime in the experiments?

- Also, how exact is the combinatorial linear sum assignment problem? Does it always solvable?

- I suggest bring the objective of CoLLike and the combinatorial linear sum assignment problem to the main paper. Those should be treated in the main body.

- It seems that there are no direct comparison between CoLLike and ELBO in the experiment section. Does it due to that the ELBO does not have tractable likelihood? Can't it be approximated by Monte Carlo estimation, for example, in [1,2]?

- Could you provide more information about the experiment setting in Section 6.2?

- In data generation perspective, how ones can generate data from the latent prior p(z)? Could you explain the sample generation process?

- I suggest authors to have proof-reading from the natives.

[1] Stick-Breaking Variational Autoencoders, https://openreview.net/pdf?id=S1jmAotxg

[2] Dirichlet Variational Autoencoder, https://www.sciencedirect.com/science/article/pii/S0031320320303174



**Strength And Weaknesses:**

The experimental results show the efficacy of the suggested CoLLike.

However, it seems that the computational complexity of O(N^3) can be problematic, where N indicates the number of data instances.


**Summary Of The Paper:**

This paper proposes Complete Latent Likelihood (CoLLike) which utilizes permutation for matching in the latent domain. The authors argues that permuting the latents and then matching to inputs leads to expressive models with exact likelihood and no posterior collapsing, compared to MaL. Also, the suggested CoLLike can link likelihood and optimal transport.


**Summary Of The Review:**

The paper is fairly-written in general. However, the authors are missing a critical point, which is an analysis on the computation complexity. Following from the authors, the provided algorithm has O(N^3) complexity, where N indicates the number of data instances. My major concern for this paper is that the utility of the proposed method may vary with computational time. Also, the clarity of the paper needs be improved. Therefore, at this moment, I'm leaning to the negative side on this paper.

---

> ### Author Response · Authors · 2022-11-19
> **Response**
>
> Thank you for the review.
>
> > At the beginning of Section 2, the authors should mention that x is an input and y is a label to be predicted, even though it is convention. Or, am I wrong?
>
> it can be treated as a convention, but we added a description for clarity
>
> > I wonder why the authors represent the latent variable z as a “label”. Does the latent z is limited to some attributes, for example? Can’t it be in any shape? It would be better to provide a graphical notation of x, y, and z, together.
>
> In the context of Complete Likelihood, $z$ can be treated as a label. As a latent variable $z$ can be anything with any shape. Like a label in the dataset.
>
> > Sampling from the prior ensures the first requirement, … -> probably (x_N, y_{pi(N)}) ?
>
> values of $z_{\pi(n)}$ creates an empirical distribution that can be
>
>
> > Figure 1 is very hard to follow. What are the circles indicate? Where is permutation pi in the figure?
>
> - we added a description
>
> > The python-like pseudo-code should be written with more consistency.
>
> this part is completely rewritten for clarity
>
> > The complexity of CoLLike seems to be a problem. Even though the authors utilized the combinatorial linear sum assignment problem with complexity of O(N^3), the problem is that N is the number of data instances which can be a huge number. Could you compare the wall-clock runtime of CoLLike against the baselines? Also, I’m curious about the effect of additional techniques: minibatch assignment, low-dimensional discrete latents, and factorized conditional. How much do they reduce the wall clock runtime in the experiments?
>
> > Also, how exact is the combinatorial linear sum assignment problem? Does it always solvable?
>
> The algorithm is exact, the problem is always solvable, for some inputs there are multiple solutions (the algorithm return only one).
>
> > I suggest bring the objective of CoLLike and the combinatorial linear sum assignment problem to the main paper. Those should be treated in the main body.
>
> We added extra description of the combinatorial complexity in the algorithm section, however, unfortunately, we were not able to bring appendices information to the body due to the page limit
>
> > It seems that there are no direct comparison between CoLLike and ELBO in the experiment section. Does it due to that the ELBO does not have tractable likelihood? Can't it be approximated by Monte Carlo estimation, for example, in [1,2]?
>
> CoLLike and ELBO can be compared, however, ELBO is only an approximation of MaL. Hence, it might be hard to tell whether some effect comes from the approximation or from the original objective. Definitely, we can use all available tools to estimate it, like MCMC and Variational Inference. This
>
> > Could you provide more information about the experiment setting in Section 6.2?
>
> We added graphical descriptions of the proposed architectures, along with a description of the exact likelihood evaluation in the algorithm section
>
> > In data generation perspective, how ones can generate data from the latent prior p(z)? Could you explain the sample generation process?
>
> Prior is assumed to be simple and easy to sample, like uniform or Gaussian. We obtain a sample $z$ and pass it to the model $p_\theta(x|z)$ to produce $x$ either in autoregressive (AR) fashion (for AR model) or using the ordinary procedure used in Normalizing Flows (for Glow models).

---

### Official Review · Reviewer_V5Wu · 2022-10-24

**Confidence:** 4
**Correctness:** 1
**Technical Novelty And Significance:** 3
**Empirical Novelty And Significance:** 3
**Recommendation:** 1

**Clarity, Quality, Novelty And Reproducibility:**

The authors approach a fundamental question of what learning means, but unfortunately the paper is written in quite a confusing manner from a narrow perspective. I would assume a paper proposing a replacement for standard learning objectives to analyse the problem from a proper statistical perspective with rigorous mathematical treatment, rather than talking about vague concepts like "the target conditional". It seems that the motivation for the work comes from specific problems (posterior collapse etc) in specific types of latent variable models and the proposed objective indeed may be a good heuristic for addressing these specific aspects, but the paper suggests it would be a general solution for learning any latent variable models. From this perspective it is highly questionable and difficult to read.

Already the starting point is odd. The paper proposes a specific kind of a point estimate for $z$ as an alternative for marginal likelihood (that integrates over $z$), rather than relating it to standard point estimates that would be much the natural baseline for this. This discrepancy remains throughout the motivation, model description and experiments, where (unsurprisingly) metrics that favor point estimates are shown to be better. The paper would be easier to understand and follow if it focused on explaining the effect of the specific constraint for $z$ (namely that the set of all $z$ must be compatible with a sample drawn from the prior), rather than attempting to describe the objective as alternative to marginalising over the latent variables.

It is good that the authors clearly mention the main assumptions, but I have trouble accepting either one of them in general case. First of all, why should the "marginal of the target conditional in the latent domain" (sic) correspond to the prior $p(z)$? The standard practice in latent variable models is to use priors that are broad to allow for the data define the actual posterior, and models with no posterior concentration would be considered poorly defined. I understand the need to avoid extreme posterior collapse in flexible deep models, but nevertheless we should typically expect $p(z|x)$ to differ quite notably from $p(z)$. You later show empirically that you indeed reach $p(z|x)=p(z)$ which nicely shows the algorithm works well and that you can satisfy the assumption, but I would consider that as negative result from the perspective of learning a useful model for the data.

Similarly, I do not understand the assumption of confidently assigning a single $z$ to each $x$. Quite contrary, I would expect $p(z|x)$ to be broad (assign clearly non-zero probability for non-diminishing set of $z$) for many common use-cases for latent variable models. For clustering, we need probabilistic models exactly when the clusters overlap severely, in recommender engines we use latent variable models because for new users we know we cannot determine $z$ accurate before seeing enough data etc. In brief, the whole assumption seems to contradict the most common use cases for the model family. Even if I somehow misunderstood the statement or we narrow down to some specific sub-family of latent variable models where this assumption makes sense, the justification 'as in any real-world dataset' is nowhere near adequate. Finally, I do not see how 'assignment of a single $z$ to each $x$' would assure the second assumption. It certainly assigns one $z$ for each $x$, but does not say anything about the actual $p(z|x)$ in the sense that we could confidently tell for future $x$ what the latent variable is. Then again, the proposed approach does not even seem to define $p(z|x)$ outside the training instances.

The presentation of the algorithm is poor; you only provide a pseudo-code rather than explaining it mathematically. As you anyway provide code that likely includes similar parts, it would be better to describe the method properly in the main paper. Furthermore, the discussion of possible speedups remains on such a high level that we do not even know whether you have derived the details (and used these in the experiments) or whether these are just ideas for future work.

The empirical experiments are similarly vague. It seems you use 'MaL' to refer to a solution of some approximation (which seems to vary from case to case) and explain the methods on a very high level ("we use Glow-like normalising flow"). I know the details are provided in the Supplement, but the presentation in the main paper is still too simplified. As a practical example, your experiments do not say anything about the computational cost of the approach. You clearly mention in technical sections that the method has high cost because of the permutation and propose a few techniques for addressing it, but you do not even say whether those were used and you do not say anything about the actual cost. Now the reader has no way of knowing whether the MaL and CoLLike are approximately as fast, or whether one of them is 1000 times slower. For a method with known computational issues you absolutely need to say something about the cost in the empirical section.

The empirical results are good in the sense that they confirm the learning algorithm, but do not tell much about the specific formulation. Replacing integration over $z$ with point estimates indeed increases MI between the estimate and the ground truth, due to reduced uncertainty, and this is what you see. You would most likely get a similar result when comparing the solution of k-means and the posterior of a probabilistic mixture, but it would not imply the former is a better objective.

Questions and minor remarks:
- What is the exact relationship to EM? It sounds like the most closely related method, rather than MaL, yet it is only briefly mentioned. Should there perhaps be some comparison as well? Even EM is 'more advanced' than your method as it approximately integrates over $z$ rather than finding a point estimate, but still a better match for comparisons.
- Does your learning algorithm propagate gradient information through LAP? Not easy to figure out from the pseudocode as it may depend on how LAP is implemented.
- It sounds like your approach works in 'transductive' setting, in the sense that you assume all $x$ to be available at training time. Can you still estimate $z$ for new samples in an inductive manner? If yes, how? This should be discussed.
- How would the model work in a case where the true $p(z|x)$ are very broad (e.g. a clustering case where the clusters are almost -- or perhaps completely -- overlapping)? Would it still find a good objective value? I strongly feel that illustrating the model in some simple examples, in addition to the main goal of flexible latent-variable models, would help clarifying the key concepts. For instance, a simple mixture model case would show the readers how the optimal solution is similar to what k-means would produce, in the sense that it is a hard clustering where the global optimum is one that assigns each sample to the closest cluster (unless I'm mistaken). An example like this would show what the effect of sampling the set of $z$ from the prior is in practice.
- How would the method work if the prior is way off? For instance, if you assume Dirichlet prior in clustering with constant cluster sizes but the data strongly suggests some clusters have ten times more samples?
- Constrained K-means does have a probabilistic interpretation; see e.g. Jitta and Klami "On controlling the size of clusters in probabilistic clustering" (AAAI, 2018).


**Strength And Weaknesses:**

Strengths:
- The paper focuses on a fundamental learning problem and hence has potential for significant impact if the contribution is considered valuable by the community
- The core idea of constraining the possible values for the latent variables based on a sample from the prior is interesting
- The learning algorithm seems to work well, in the sense that we observe empirically the properties we should expect from the objective (higher MI, lower KL between the prior and the conditional etc)

Weaknesses:
- The objective is poorly motivated, using only high-level argumentation like posterior collapse in some specific latent-variable models as justification for limitations of marginal likelihood, and it relies on possibly questionable assumptions (sharp distribution for the latents). The main story focusing on its relationship with marginal likelihood is odd.
- The presentation is on a too high level. For instance, the main method is described only in terms of pseudo-code and the experiments are near impossible to parse from the main paper, for instance to figure out how the proposed method was used in these.
- The computational cost of operating with permutations is dismissed lightly; the authors mention it as a challenge and outline possible solutions on a high level, but do not transparently communicate what it means in practice. The experiments completely omit all discussion on the computational cost or even whether the proposed speedups were used.
- The notation and terminology are confusing and non-standard, making the paper difficult to read. For example, the phrases "unknown $z|x$ target conditional" or "target $z|x$ distribution" used in Introduction do not mean anything at that stage. Similarly, Section 3 is written in rather confusing terminology and makes standard concepts sound unjustified choices; the choice of conducting learning in a manner that is faithful to the modelling assumptions is called "keeping the model posterior unchanged" etc.

**Summary Of The Paper:**

The authors propose a new learning objective for fitting latent variable models. The basic idea is to sample a set of latent variables from the prior and find a permutation of those that aligns well with the observed data. The method provides a point estimate of the latent variables, but the paper is written from a perspective where it is suggested as an alternative to marginalising over the latent variables rather than focusing on the effect of the constraint imposed over standard joint likelihood. The approach is demonstrated primarily in context of supervised classification problems, where it is shown to find latent variables that align well with known classes.

**Summary Of The Review:**

A bold attempt of re-defining the fundamental learning objective of latent variable models, but there are too many open questions regarding the theoretical foundations and also the empirical side. The biggest issue is that the approach is compared against marginal likelihood when it actually corresponds to a specific way of constraining a point estimate for the complete likelihood, and because of this mis-positioning the paper actually avoids discussing what the effect of sampling the set of possible latent variables from the prior is in practical cases or even in easy examples. From the perspective of general latent variable models the assumptions made for deriving the new objective are either shaky or even wrong, but it may be that for specific types of models they provide reasonable inductive biases. However, for that the paper would need to be re-written from a very different perspective.

---

> ### Author Response · Authors · 2022-11-19
> **Response**
>
> Thank you for the review.
>
> The critics in the review rely on following considerations:
> - CoLLike objective is not justified, unlike Marginal Likelihood.
>     - This is not correct since the Marginal Likelihood is not justified for representation learning.
> - High-level argumentation
>     - Every term stated in the review as a "high level" or "vague" has a precise mathematical counterpart and cannot be considered in this terms.
> - Wrong assumptions
>     - We reworked Section 2 to make the statement clearer. First assumption results in an empirical distribution, that maximizes mutual information. Both broad and sharp distributions can be described by empirical distributions. The second requirement is natural if the prior is assumed to be known. Moreover, these requirements realize Infomax principle as we also added to the paper.
> - The results are expected
>     - Most of the published works present expected results. Expected results indicate that we are not missing something.
> - Effects of sampling
>     - Effects of sampling are minor in the considered cases as can be seen from the low $KL$ divergences. However, this can be a significant problem for high-dimensional $z$. We added this consideration to the objective section.
> - The objective should be considered as a point estimate
>     - "Point estimate" (of the posterior we guess) is a convenient perspective for matching distributions of the observed variables. It makes no sense from the representation learning perspective.
> - The algorithm and the model description
>     - We reworked this part significantly.
>
> Answers to questions:
>
> > What is the exact relationship to EM? It sounds like the most closely related method, rather than MaL, yet it is only briefly mentioned.
>
> We clarify the connection in the paper, the relationship is in the alternating maximization.
>
> > Should there perhaps be some comparison as well? Even EM is 'more advanced' than your method as it approximately integrates over
> rather than finding a point estimate, but still a better match for comparisons.
>
> EM is not needed in our case since the exact likelihood is available.
>
> > Does your learning algorithm propagate gradient information through LAP? Not easy to figure out from the pseudocode as it may depend on how LAP is implemented.
>
> No, we made this clear in the reworked Algorithm section.
>
> > It sounds like your approach works in 'transductive' setting, in the sense that you assume all to be available at training time. Can you still estimate for new samples in an inductive manner? If yes, how? This should be discussed.
>
> We can use variational inference as a common tool for this as well as CoLLike algorithm itself. We added it to the algorithms section.
>
> > How would the model work in a case where the true are very broad (e.g. a clustering case where the clusters are almost -- or perhaps completely -- overlapping)? Would it still find a good objective value? I strongly feel that illustrating the model in some simple examples, in addition to the main goal of flexible latent-variable models, would help clarifying the key concepts. For instance, a simple mixture model case would show the readers how the optimal solution is similar to what k-means would produce, in the sense that it is a hard clustering where the global optimum is one that assigns each sample to the closest cluster (unless I'm mistaken). An example like this would show what the effect of sampling the set of from the prior is in practice.
>
> We are interested in "useful representations" that can be clearly verified. Supervised datasets are best for this purpose. Datasets with broad distributions are not common.
>
> > How would the method work if the prior is way off? For instance, if you assume Dirichlet prior in clustering with constant cluster sizes but the data strongly suggests some clusters have ten times more samples?
>
> This is interesting. We need to introduce learnable prior. This is a direction for future work.
>
> > Constrained K-means does have a probabilistic interpretation; see e.g. Jitta and Klami "On controlling the size of clusters in probabilistic clustering" (AAAI, 2018).
>
> We add this to the paper.
>
> ----
> Further discussion:
>
> > The objective is poorly motivated, using only high-level argumentation like posterior collapse in some specific latent-variable models as justification for limitations of marginal likelihood
>
> All the concepts referred in the review as "high-level" (including "target conditional" and "posterior collapse") comes with a precise mathematical counterpart, which cannot be addressed as "high-level".
>
> > The main story focusing on its relationship with marginal likelihood is Marginal Likelihood is the most common objective to learn latent variable models, this motivates the choice. It is unclear why this comparison is "odd".odd.
>
> Marginal Likelihood is the most common objective to learn latent variable models, this motivates the choice. It is unclear why this comparison is "odd".

---

> > ### Author Response · Authors · 2022-11-19
> > **Response**
> >
> >
> > > The presentation is on a too high level. For instance, the main method is described only in terms of pseudo-code and the experiments are nearly impossible to parse from the main paper, for instance to figure out how the proposed method was used in these.
> >
> > We significantly reworked this part to make this part more clear.
> >
> > > The computational cost of operating with permutations is dismissed lightly; the authors mention it as a challenge and outline possible solutions on a high level, but do not transparently communicate what it means in practice. The experiments completely omit all discussion on the computational cost or even whether the proposed speedups were used.
> >
> > This part is clarified in the paper and addressed in the comment [[link](https://openreview.net/forum?id=hO8qWILpJ3J&noteId=onCERPld5Z)]
> >
> > > The notation and terminology are confusing and non-standard, making the paper difficult to read. For example, the phrases "unknown $z|x$
> > target conditional" or "target $z|x$ distribution" used in Introduction do not mean anything at that stage.
> >
> > We see no option to confuse these cases with anything, $KL$ between the model distribution and other distribution leaves no options to interpret it in some other way. Introduction of these terms is motivated by the perspective proposed in the paper.
> >
> > > Similarly, Section 3 is written in rather confusing terminology and makes standard concepts sound unjustified choices; the choice of conducting learning in a manner that is faithful to the modelling assumptions is called "keeping the model posterior unchanged" etc.
> >
> > The choice of the Marginal Likelihood is not justified for representation learning. If we "conducting learning in a manner that is faithful to the modelling assumptions" for any expressive model, we experience posterior collapse with high probability.
> >
> > > I would assume a paper proposing a replacement for standard learning objectives to analyse the problem from a proper statistical perspective with rigorous mathematical treatment, rather than talking about vague concepts like "the target conditional".
> >
> > Marginal Likelihood (standard learning objective) is rigorously justified only for learning distribution of the observed variables. We find counterproductive the fact that it is standard and treated as justified for representation learning. We use common statistical tools like $KL$ and mutual information to perform analysis. The "target conditional" concept has a precise mathematical counterpart, therefore, it cannot be vague.
> >
> > > It seems that the motivation for the work comes from specific problems (posterior collapse etc) in specific types of latent variable
> >
> > The problems are not specific, informativeness is the fundamental property of any useful representation. The types of the latent variable models are not specific. Factorized distributions is only a small subset of a distributions family with no factorization assumption.
> >
> > > suggests it would be a general solution for learning any latent variable models
> >
> > We clearly state in the first sentence of the abstract that it is an **alternative** to MaL
> >
> > > Already the starting point is odd. The paper proposes a specific kind of a point estimate for as an alternative for marginal likelihood (that integrates over ), rather than relating it to standard point estimates that would be much the natural baseline for this.
> >
> > "Point estimate" (of the posterior we guess) is a convenient perspective for matching distributions of the observed variables. It makes no sense from the representation learning perspective.
> >
> > > rather than attempting to describe the objective as alternative to marginalising over the latent variables
> >
> > Motivation of the objective comes from problems of the Marginal Likelihood. Furthermore, this objective is widely used in representation learning.

---

> > ### Comment · Reviewer_V5Wu · 2022-11-21
> > **Minor comments**
> >
> > I acknowledge that I have read all the responses you provided, but do not see reasons to address all of them directly. However, I would like to comment on a few:
> >
> > > We are interested in "useful representations" that can be clearly verified. Supervised datasets are best for this purpose. Datasets with broad distributions are not common.
> >
> > Supervised datasets may indeed be best for technical evaluation, but are you really saying that you are intending the whole method to only be used in cases where the data corresponds to some sort of clearly separated groups (classes)? Sounds like a fairly limited use-case or a strong assumption to be made.
> >
> > Note that my comment was not really about asking your to evaluate it on that kind of cases but to provide some simple conceptual illustrations that would better help the reader to understand how the method works. I feel that it would be easier to do that in e.g. two-dimensional cases where we would immediately see how the method enforces discrete latent variables even if not supported by the data, how changing the prior distribution changes the result etc. I'm not saying these would be necessarily bad properties as such, but communicating their effects better for the readers would be important.
> >
> > > > How would the method work if the prior is way off? For instance, if you assume Dirichlet prior in clustering with constant cluster sizes but the data strongly suggests some clusters have ten times more samples?
> >
> > > This is interesting. We need to introduce learnable prior. This is a direction for future work.
> >
> > I agree with your choice of words ("we **need** to") because this indeed feels like a significant limitation in the current solution. Your approach would currently not work at all e.g. in simple clustering tasks (unless the practitioner started tweaking priors to be compatible with what the inference algorithm assumes), because exchangeability results in a priori equal cluster sizes but we never really expect it to hold a posteriori. I know that this kind of tasks are not the main focus of your work, but completely failing to solve a simple task that is one of the easiest examples of latent variable models is still an alarming sign about possible issues also in more complex models. Even if you only focused on cases where this is not an issue, it would still be critically important to explain for the reader the ways the approach could fail and explain why they are not a problem in the cases you care about.
> >
> > A learnable prior sounds like a possible way to resolve this, but does not sound trivial since your method is motivated directly by matching the conditional with the prior. If you make the prior learnable you might need to change the story as well, and maybe the choice of the learning objective for the prior is not obvious either.

---

> > > ### Author Response · Authors · 2022-11-29
> > > **Response**
> > >
> > > Thank you, we appreciate your impact on our work.
> > >
> > > The case of not clearly separated classes can be modeled by the target empirical distribution. In this case, some close points can be assigned to different classes. This kind of example might be very illustrative. In the simplest case, we can imagine constrained K-Means applied to such data. This corresponds to factorized $p(x|z)$ and some categorical prior $p(z)$ as mentioned in the paper. The idea of providing an illustration for two-dimensional cases is undoubtedly a good one. We thought about constrained K-Means as such illustration but this case does not cover the problems we address in the paper. If we use a powerful $p_\theta(x|z)$ the interesting part is in the learning dynamics, which is challenging to visualize in a static picture. We could use a model with reverse factorization, however, the work is devoted to regular factorization. Nevertheless, we think that there might be another good and simple example.
> > >
> > > About the learnable prior. We see at least a couple of approaches that can be used to introduce a learnable prior. In both cases, the objective remains close to the original one. There are two parts of the model that need to be modified for learnable marginal distribution in the $z$ domain. Using the $KL$ form of the objective $KL(p_\delta(x) p_\pi(z|x) || p_\theta(x|z) p(z))$ we see that we need flexibility in both the model marginal $p(z)$ and in the target marginal $p_{\delta\pi}(z) = \int p_\delta(x) p_\pi(z|x) dx$. Making the model part learnable is straightforward and represented in the literature. To make $p_{\delta\pi}(z)$ flexible we can sample more than $N$ values from the prior. The assignment problem is solved seamlessly for rectangular cost matrices. For instance, we can sample $2 N$ values from the prior and pick only $N$ of them as targets using the objective. Another way to introduce flexibility to the target distribution is to apply local optimization to the sampled $z$ values. For instance, we can use gradient techniques to update continuous $z$ values and local (e.g. in terms of Hamming distance) search for discrete ones.

---

### Official Review · Reviewer_G4Xy · 2022-11-02

**Confidence:** 3
**Correctness:** 4
**Technical Novelty And Significance:** 4
**Empirical Novelty And Significance:** 4
**Recommendation:** 8

**Clarity, Quality, Novelty And Reproducibility:**

quality: fundamental contributions with sound justifications.

clarity: presentation is clear

originality: novel

**Strength And Weaknesses:**

Strength:
This paper address a fundamental problem in all machine learning training and it potentially has a big impact.
Many connection and differences with existing objective are discussed - it was enlightening.

Weakness:
Some concern on the permutation estimation exist due to inefficiency.


Comments:
- " target conditional is actually specified and equals $p_{\theta} (z|x)$ if we ask what distribution we want to mimic":  the exact form of $p_{\theta} (z|x)$ is still not specified, and it could be anything since it canceled out. If $p_{\theta} (z|x)$ is changed to any other distribution,  MaL  remains the same. This seems to contradict the statement.
- " the CoLLike objective allows learning models with reverse factorization": ELBO has been used in VAE like structure, which resembles the function form. In addition, it would be good to provide examples on advantage or necessity of reverse factorization.
- some references are not cited correctly. e.g., " InfoMax principle Huszar (2017))".
- Eq 3: does the permutation mean all possible label assignments for all samples? How is it a permutation if the number of each class within Z are different?
- "since the set of all possible π values is countable and finite, optimization can be performed by an exhaustive search." it seems for large sample size N, this is not truly feasible with exhaustive search. Even with Hungarian complexity O(N^3), it is still quite a big number, in particular in large datasets popular nowadays.
- Hence, it would be interesting to see the time comparison between different objectives in the experiments.
- " we emphasize our focus on learning useful z|x instead of simplifying the model with factorized x|z conditional" can authors elaborate on this point? what if x|z is not factorized?
- "To reduce the high variance of CoLLike unsupervised classification": authors should have discussed this point earlier of this work.

**Summary Of The Paper:**

This paper proposes a complete latent likelihood (CoLLike), an alternative to the marginal likelihood (MaL) objective that is standard in ML algorithms (MaL) today. A systematic comparison between COLLike and Mal in the KL framework is compared. By discussing posterior collapse and high deviation of the aggregated posterior from the prior, authors show CoLLike can prevent these drawbacks. Empirical studies on image and NLL tasks show consistent improvement of CoLLIke over MaL.

**Summary Of The Review:**

a fundamental and potentially impactful work to address the training objective in ML algorithms today.

---

> ### Author Response · Authors · 2022-11-09
> **Response to comments**
>
> Thank you for the review, we appreciate your feedback.
>
> - > " target conditional is actually specified and equals $p_\theta(z|x)$ if we ask what distribution we want to mimic": the exact form of $p_\theta(z|x)$ is still not specified, and it could be anything since it canceled out. If is changed to any other distribution, MaL remains the same. This seems to contradict the statement.
>     - After we define the model joint as $p_\theta(x, z) = p_\theta(x | z) p(z)$ the value of $p_\theta(z|x)$ is defined but not specified in an easily accessible way, namely, we need integration to get the $p_\theta(x)$ to later use it in the Bayes' theorem $p_\theta(z|x) = p_\theta(x, z) / p_\theta(x)$. The ELBO objective is very illustrative in this case: if we minimize ELBO only with respect to variational parameters $\phi$ of the distribution $q_\phi(z|x)$, we make $q_\phi(z|x)$ closer to $p_\theta(z|x)$ (a general case is in Eq. 4). Then, you precisely point at the core problem of the MaL objective: it is irresponsible to the $p_\theta(z|x)$ and care only about $p_\theta(x)$. This is because MaL is constructed to match distributions only in the visible domain, and all properties of $p_\theta(z|x)$ are neglected as inessential ones. However, if we imagine that this is an objective for obtaining useful joint distribution $p_\theta(x, z)$ but not only $p_\theta(x)$ we see that we pursue constancy of the $p_\theta(z|x)$, which results in cancellation of $z|x$ parts of the objective. So, if we are interested in matching data distribution $p(x)$, we can easily cancel out the $p_\theta(z|x)$ and do not bother about it, in turn, if we are looking for relation of $x$ and $z$ represented by the model joint we need to investigate what joint is our target, and in this case, cancellation moves us to the point when we cannot say anything about joints.
> -  > " the CoLLike objective allows learning models with reverse factorization": ELBO has been used in VAE like structure, which resembles the function form. In addition, it would be good to provide examples on advantage or necessity of reverse factorization.
>     - The target joint in the ELBO case indeed a distribution with a reverse factorization (relative to the common $p_\theta(x|z) p(z)$ one). In some sense, during optimization of ELBO, we optimize the joint $p_\delta(x) q_\phi(z|x)$, which has reverse factorization) to be similar to the model joint $p_\theta(x, z)$.
>     - Reverse factorization itself is an another inductive bias which can be both fruitful or not in terms of informativeness of the learned model. Some works in the Connections section successfully use it. Moreover, the reverse factorization models potentially can be much faster and lighter if we assume that the $p_\theta(x)$ part of the $p_\theta(x) p_\theta(z|x)$ is uniform. This is an interesting direction, it will be added to the paper.
>  - > some references are not cited correctly. e.g., " InfoMax principle Huszar (2017))".
>     - This will be fixed
> - > Eq 3: does the permutation mean all possible label assignments for all samples? How is it a permutation if the number of each class within Z are different?
>     - Not exactly all possible label assignments but only those that are produced by rearrangements of the fixed collection sampled from the prior $\{ z_1, ..., z_N \}$. For instance, if we have a dataset $\{x_1, x_2}$ we sampled three values from some prior $\{z_1, z_2\}$. These sequences are completely described by four numbers/vectors. Then, all permutations will be: $\{(x_1, z_1), (x_2, z_2) \}$ and ${(x_1, z_2), (x_2, z_1) \}$.
>
> - > "since the set of all possible π values is countable and finite, optimization can be performed by an exhaustive search." it seems for large sample size N, this is not truly feasible with exhaustive search. Even with Hungarian complexity O(N^3), it is still quite a big number, in particular in large datasets popular nowadays.
>     - definitely, the exhaustive search has limited applicability, we will reformulate this part.
> - > Hence, it would be interesting to see the time comparison between different objectives in the experiments.
>     - this is an important question and we refer to [our comment](https://openreview.net/forum?id=hO8qWILpJ3J&noteId=PwYO71tx4qz) posted out of this discussion branch.
> - > " we emphasize our focus on learning useful z|x instead of simplifying the model with factorized x|z conditional" can authors elaborate on this point? what if x|z is not factorized?
>     - We believe that this question is highly related to [our reply](https://openreview.net/forum?id=hO8qWILpJ3J&noteId=1FhcXdyUfD).
> - > "To reduce the high variance of CoLLike unsupervised classification": authors should have discussed this point earlier of this work.
>     - We refer to the variance of the accuracy from the previous section. This will be clarified.

---

### Author Response · Authors · 2022-11-09
**General Appeal to the Reviewers**

We want to thank the reviewers for the feedback and address this message to all of them because we find the following material especially important. Nevertheless, it is only implied in the present work and should be incorporated into it.

We want to share our view on the history of Marginal Likelihood (MaL) as a representation learning algorithm. MaL can be used for two distinct tasks:
1. learning models of the data distribution $p^*(x)$, where data instances $x$ can contain labels or not. In this case, we are interested in models of the form $p_M(x) = \int p_M(x, z) dz$.
2. learning representations, where we are interested in $p_M(z|x)$.

The history of the development of MaL is devoted to the first task. For this task, the entire focus is on capturing the true underlying data distribution $p^*(x)$ by the model $p_M(x)$, while the exact form of $p_M(z|x)$ is out of interest. The only requirement for $p_M(z|x)$ is to produce $p_M(x)$ that matches $p^*(x)$. MaL is not only justified for this task by $KL(p_M(x) ||p^*(x))$ [1, sec. 4.2.2] but can also be considered as a general and rigorously founded approach (rather than ad hoc) for learning distributions of observed variables because it realizes Occam’s razor [2, 3 ch. 22].

Before the deep learning era, modelling complex multimodal $p^*(\mathbf{x})$ with high-dimensional $\mathbf{x}$-s was widely approached by latent variable models with factorized $p_M(\mathbf{x}|\mathbf{z}) =  p_M(x_1|\mathbf{z}) p_M(x_2|\mathbf{z})...=\prod_i p_M(x_i|\mathbf{z})$ and MaL. This was mainly dictated by the computational reasons since factorized distributions are both easy to compute and easy to approach mathematically. For such models, it is impossible not to utilize $\mathbf{z}$, since independent $\mathbf{x}$ and $\mathbf{z}$ resulting in $p_M(\mathbf{x}|\mathbf{z}) =  p_M(\mathbf{x})$ is too simple. Hence, such models must learn some relations between $\mathbf{x}$ and $\mathbf{z}$, if it is possible to utilize it to increase the similarity between distributions in the $\mathbf{x}$ domain. Researchers in this field noticed that these relations are often coherent with people's needs, like, for instance, for certain tasks $\mathbf{z}$ becomes correlated with labels without utilization of labels at all. And this looks like a "holy grail", getting annotations without human labor. Therefore, useful representations (task 2 in the list above) were an encouraging side effect of the MaL application, because it was constructed to make good models only in the $\mathbf{x}$ domain.

Due to this surprising success of MaL in learning useful representations (relations between $x$ and $z$ represented by $p_M(x, z)$) researchers begin to use it as a tool to obtain valuable $p_M(z|x)$, although it is out of interest in the original problem statement (match $p_M(x)$ and $p^*(x)$). As a result, MaL became treated as a tool suited for representation learning.

However, in the deep learning era, things changed. Researchers developed powerful models with a non-factorized distribution like autoregressive or flow-based ones. These models can capture complex distributions without latent variables. Of course, it was really exciting to try those models in a latent variable setting to possibly get even better representations. However, the result of using powerful models extended by latent variables was frustrating [4], the model learns not to relate $z$ to $x$ and the resulting model distribution is such that the posterior of every $x$ equals prior. This is commonly referred to as 'posterior collapse' and occurs in a variety of generative models (see the links in the paper). However, MaL is still recognized as a tool for representation learning. To support this fact, we provide a citation from [5]:
> Recent work in unsupervised representation learning has focused on learning deep directed latent-variable models. Fitting these models by maximizing the marginal likelihood or evidence is typically intractable, thus a common approximation is to maximize the evidence lower bound (ELBO) instead. However, maximum likelihood training (whether exact or approximate) does not necessarily result in a good latent representation, as we demonstrate both theoretically and empirically.

The representation learning task still lacks a grounded objective. We find it extremely important to search for a general and well-founded objective for learning representations, as well as to encourage researchers to explore the methods of building such objectives. Nevertheless, we understand that our work may seem an ad hoc solution. It was indeed originally developed as an ad hoc one; however, the need of an ad hoc solution implies that the choice of the original objective was not justified. We discovered a way from first principles to this ad hoc solution, which can be used directly or modified to approach task 2. Now this objective can be used as justified. And the discovery of this way was astonishing to us.

---

> ### Author Response · Authors · 2022-11-09
> **Bibliography**
>
> [1] Kevin P. Murphy.Probabilistic Machine Learning: An introduction. MIT Press, 2022. URL: probml.ai.
>
> [2] Jeffreys, H.The theory of probability. The Claren-don Press, Oxford, 1939.
>
> [3] MacKay, D. J. Information theory, inference and learning algorithms. Cambridge University Press, 2003.
>
> [4] Samuel R. Bowman, Luke Vilnis, Oriol Vinyals, Andrew M. Dai, Rafal J ́ozefowicz, and Samy Bengio. Generating sentences from a continuous space. In Yoav Goldberg and Stefan Riezler (eds.),Proceedings of the 20th SIGNLL Conference on Computational Natural Language Learning, CoNLL 2016, Berlin, Germany, August 11-12, 2016, pp. 10–21. ACL, 2016
>
> [5] Alexander A. Alemi, Ben Poole, Ian Fischer, Joshua V. Dillon, Rif A. Saurous, and Kevin Mur-phy. Fixing a broken ELBO. In Jennifer G. Dy and Andreas Krause (eds.),Proceedings of the35th International Conference on Machine Learning, ICML 2018, Stockholmsm ̈assan, Stockholm,Sweden, July 10-15, 2018, volume 80 of Proceedings of Machine Learning Research, pp. 159–168. PMLR, 2018
>
> [6] Yang You, Jonathan Hseu, Chris Ying, James Demmel, Kurt Keutzer, and Cho-Jui Hsieh. Large-batch training for lstm and beyond. In Proceedings of the International Conference for High-Performance Computing, Networking, Storage and Analysis, SC ’19, New York, NY, USA, 2019.Association for Computing Machinery.
>
> [7] Yang You, Jing Li, Sashank J. Reddi, Jonathan Hseu, Sanjiv Kumar, Srinadh Bhojanapalli, Xiao-dan Song, James Demmel, Kurt Keutzer, and Cho-Jui Hsieh. Large batch optimization for deep learning: Training BERT in 76 minutes. In 8th International Conference on Learning Representations, ICLR 2020, Addis Ababa, Ethiopia, April 26-30, 2020.
>
> [8] Chen Xing, Devansh Arpit, Christos Tsirigotis, and Yoshua Bengio. A walk with SGD. CoRR,abs/1802.08770, 2018. URL http://arxiv.org/abs/1802.08770.

---

> ### Author Response · Authors · 2022-11-09
> **Computational Complexity**
>
> Many reviewers raised questions about computational complexity and consequent practical applicability, which are undoubtedly important. They are definitely missing in the paper and will be added.
>
> Quick summary: in all our experiments, training a single model requires fairly modest resources: a single GPU and approximately a single day and does not require hundreds of TPUs to be reproduced or applied to another domain. The Hungarian $\mathcal{O}(B^3)$ ($B$ - batch size) complexity is not a problem for $B$ up to thousand. Computation of likelihood for all pairs increases the compute time (compared to the supervised setting) proportional to $K$ for low-dimensional discrete setting and to $B$ for high-dimensional and continuous settings ($K$ - number of classes).
>
> Going to particular timings.
> CIFAR (batch_size = 64):
> - CoLLike: 0.44 s per batch
> - MaL: 0.63 s per batch
> - Supervised: 0.21 s per batch
> AG News (batch_size = 64):
> - CoLLike: 0.102 s per batch
> - MaL: 0.038 s per batch
> - Supervised: 0.014 s per batch
> Where supervised corresponds to the same architecture but with given labels and regular complete likelihood objective. We utilized “minibatch” and “low-dimensional discrete” techniques from 4.1. This allows us to compute 10 forward passes for CIFAR and 4 for AG News (compared to 1 pass in the supervised setting). Training with the same batch size but without “low-dimensional discrete” (e.g. if we have continuous latents) is definitely possible but far less pleasant. Batch sizes around 512-1024 are a serious problem, despite we believe that the naive permutation estimation we use is redundant and can be significantly improved.
>
> Towards the $\mathcal{O}(B^3)$ complexity. This part seems to raise no problem at the moment. We used batch size 64 for all experiments. The solution of the linear assignment problem takes around 300 microseconds while the forward and backward passes take hundreds of milliseconds. Thus, the optimal assignment takes orders of magnitude less time compared to the network computations. To understand the limits of the Hungarian algorithm, here are the mean timings for different sizes of the problem: 64 - 0.29 ms, 512 - 20.6 ms, 1024 - 96 ms, 2048 - 430 ms, 4096 - 1.93 s. Hence, we expect Hungarian algorithm to be applicable to batch sizes up to thousands. Nevertheless, large batch sizes are possibly not good for generalization [6, 7, 8].
>
> About the complexity of forward and backward passes of the network. In the supervised setting, we need $B$ forward passes and $B$ backward passes for a minibatch. In our experiments, for CoLLike, with the help of “minibatch” and “low-dim” we need $K \cdot B$. If we assume that the forward and backward passes have the same computational complexity, then the CoLLike algorithm takes approximately $K/2$ extra compute time for the network. In case we have $K > B$ (e.g. ImageNet) or continuous latents, we can definitely use batch sizes around $16..64$ or find a way not to compare every single pair of $x_i$ and $z_j$. Comparison of all pairs, which in addition are picked randomly, is definitely redundant.
>
> Just in case you are interested in the complexity of MaL. It requires $K \cdot B$ forward and backward passes because we not only need to compute $p_\theta(x, z)$ for every possible $z$ but also to backpropagate through the exact marginal likelihood $\log p_\theta(x) = \sum_{z'} \log p_\theta(x, z') = \log p_\theta(x, z=0) + \log p_\theta(x, z=1) + ...$ (which is possible to compute for low-dimensional $z$). This results in $K -1$ extra network compute time with the same forward-backward assumption. Extra compute is the price paid for unknown labels. As an example, for 4 classes we expect the compute time to double compared to the supervised setting (without the optimal permutation overhead).

---

> ### Comment · Reviewer_V5Wu · 2022-11-17
> **Response to appeal**
>
> Thank you for the clarification. I read your revised description and I can follow your reasoning in most parts. However, this response is quite different from the original story in the paper, and even if we assumed the technical side to be valid we would still be looking at a case where the story is simply not ready yet. Consequently, I cannot see a way how the paper would be publishable in its current form, even though I agree that some of the practical elements are interesting and potentially valuable.
>
> Some small remarks regarding the new story:
>
> > This was mainly dictated by the computational reasons since factorized distributions are both easy to compute and easy to approach mathematically. For such models, it is impossible not to utilize $z$, since independent $x$ and $z$ resulting in $p(x|z) = p(x)$ is too simple. Hence, such models must learn some relations between $x$ and $z$, if it is possible to utilize it to increase the similarity between distributions in the $x$ domain.
>
> This reasoning seems quite backwards to me. While factorising the distribution is partially motivated by computational tractability, the main reason for expressing $p(x|z)$ like that has been that we **want** to do so. This is a general design pattern that allows easily encoding various biases and modelling assumptions to aid constructing models that learn useful things. I think you actually say pretty much the same thing when you talk about "such models must learn some relations" and "researchers noticed that these are coherent with people's needs", but then somehow make it sound like it is a bad idea to design models that capture our intuitions and focus on providing information about the aspects we care about. This simply does not make sense to me.
>
> > Due to this surprising success of MaL in learning useful representations (relations between $x$ and $z$ represented by $p(x,z)$ ) researchers begin to use it as a tool to obtain valuable $p(z|x)$, although it is out of interest in the original problem statement (match $p(x)$ and $p^*(x)$).
>
> This is equally confusing. In extremely many cases the fundamental goal has always been to learn useful $p(z|x)$, so I cannot understand why you somehow interpret that as a side-result of some *'original problem statement'*. I think you are somehow confusing the technical objective (marginal likelihood) and the actual task (learning useful representations), which by no means have to match. I don't think any actual modeller would answer *'I want to match the marginal distributions'* if you ask about their actual modelling goal, but instead they would say *'I want to understand how X works'* and continue by explaining they use marginal likelihood as the standard statistical measure to ensure the model is properly grounded in the data.
>
>
>
> Overall, I recommend you to continue working on the approach, but I suggest trying to express the main claims and motivations in maximally standard language and to avoid making overly strong claims regarding fundamental principles of statistical modelling. You do not need those arguments to justify a model if it is able to learn useful representations. However, I think you still need to work also on the way you quantify the usefulness of the representations, since (as pointed in the original review) some of the measures you now use appear to be biased towards favoring your approach.
>
> Finally, thank you for opening the computational cost. It is good to hear it is not an issue in practice, and better communicating that for the reader will certainly make the paper stronger.

---

> > ### Author Response · Authors · 2022-11-29
> > **Response**
> >
> > Thank you for the detailed clarification. We appreciate your impact on our understanding of the problem.
> >
> > First of all, our investigation of the roots of the objective leads us to the conclusion that our interpretation of the historical part is indeed far from reality. Below we try to highlight our misinterpretations as well as clarify some points.
> >
> > We can treat Gauss's work on least squares [1] as a starting point (both from practical and theoretical perspectives) for the maximum likelihood objective. Gauss used maximum likelihood to estimate the parameters of trajectories of celestial bodies. He used Kepler's laws as a model and a uniform distribution as a prior. However, there is a significant difference between Gauss's case and current latent variable models. The model is constructed on the basis of physics laws and the observations almost precisely come from some joint in the family spanned by the model.
> >
> > One of the first latent variable models is presented in Spearman's work "General Intelligence" [2]. The approach used in this work can be treated as fitting a Factor Analysis model. The work is devoted to the discovery of latent factors that can be explained as intelligence. The method of fitting the model is explained in terms of correlations and its relation with likelihood is clearly non-trivial. The consequent work "On the mathematical foundations of theoretical statistics" by Fisher [3] was devoted to justification of the maximum likelihood objective using such properties as consistency, efficiency, and sufficiency. However, all these properties hold only if the data comes from the model distribution. For [1] we can say (with minor remarks) that the data comes from the model distribution, for [2] and the majority of latent variable models this assumption does not hold. This problem is addressed in  Fisher's work [3, sec. 3, "problem (1)"]. Nevertheless, we can use the value of the likelihood to determine (using $KL$) the discrepancy between the model and the data distributions on the test set.
> >
> > Further development of the objective is devoted to the application of the Bayesian framework to Maximum Likelihood objective. Jeffreys' "Theory of Probability" [4] is fundamental in this direction. This work can be considered as a foundation of the Marginal Likelihood objective. Along with the already mentioned Occam’s razor, there are many other arguments in favor of Marginal Likelihood including: less overfitting, well-sound uncertainty estimation, probabilistic reasoning, and many others.
> >
> > We emphasize that the goal of all mentioned approaches was in the parameters but not in the distribution they produce. Representations definitely were not a "side-effect".
> >
> > However, from the representation learning perspective, the story highly relies on the building models that produce distribution close to the data distribution (has a high likelihood and consequently low $KL$). If the model and the data distributions are close to each other then the parameters of the model can be treated as explanatory factors of the data and potentially useful representations. Nevertheless, there are two significant problems with this approach. Firstly, usefulness or interpretability of the learned parameters is determined by the model structure, which in turn relies on the ingenuity of the statistician. For relatively simple or low-dimensional distributions such models are widely and successfully used. However, for a variety of complex distributions such as images or texts development of models with useful parameters is challenging, which is reflected by problems of posterior collapse and mismatch in the latent domain. Secondly, the objective is aimed at explaining all factors present in data regardless of the usefulness while we are interested only in specific ones. Instead of modeling all factors that shape the data, we propose to directly search for representations that are useful to us. This is a key difference between our method and the standard statistical framework. Learning models with reverse factorization $p_\theta(x) p_\theta(z|x)$ (as in e.g. NAT) is very illustrative in this case. Letting $p_\theta(x)$ be fixed and uniform, we focus entirely on the $p_\theta(z|x)$ without modeling the data distribution.
> >
> > [1] Gauss, C. F.(1809). Theoria Motus Corporum Coelestium. Perthes et Besser, Hamburg. Translated, 1857, as Theory of Motion of the Heavenly Bodies Moving about the Sun in Conic Sections, trans. C. H. Davis. Little, Brown; Boston.Reprinted, 1963, Dover, New York.
> >
> > [2] Spearman, Charles. "E. 1904.“General Intelligence” Objectively Determined and Measured." American Journal of Psychology 15.2 (1904): 201-293.
> >
> > [3] Fisher R. A. 1922 On the mathematical foundations of theoretical statistics. Philosophical Transactions of the Royal Society of London. Series A, Containing Papers of a Mathematical or Physical Character
> >
> > [4] Jeffreys, H. The theory of probability. The Clarendon Press, Oxford, 1939.

---

### Decision · Program_Chairs · 2023-01-20

**Decision:**

Reject

**Justification For Why Not Higher Score:**

As reviewers pointed out, severaAl issues are unclear (e.g., what if the prior is not known in advance?), the presentation is to be improved, and the authors may consider including an analysis of the computation complexity of the proposed matching procedure.

**Justification For Why Not Lower Score:**

The tackled problem is fundamental, important, and interesting, and the empirical results illustrate the efficacy of the proposal.

**Metareview: Summary, Strengths And Weaknesses:**

This paper proposes a Complete Latent Likelihood (CoLLike) objective, as an alternative to commonly used marginalized likelihood objectives for learning latent variable models. The basic idea is that when the prior over latent variables are known or given, the proposed method generates i.i.d. latent variables from the prior, and find a permutation between the generated latent variables and observed samples. The tackled problem is fundamental, important, and interesting, and the empirical results illustrate the efficacy of the proposal. However, as reviewers pointed out, several issues are unclear (e.g., what if the prior is not known in advance?), the presentation is to be improved, and the authors may consider including an analysis of the computation complexity of the proposed matching procedure.